# SADDLE-TO-SADDLE DYNAMICS IN DEEP RELU NETWORKS: LOW-RANK BIAS IN THE FIRST SADDLE ESCAPE

**Ioannis Bantzis** *
EPFL
Lausanne, Switzerland
`ioannis.bantzis@alumni.epfl.ch`

**James B. Simon**
UC Berkeley and Imbue
Berkeley and San Francisco, USA
`james.simon@berkeley.edu`

**Arthur Jacot**
Courant Institute, NYU
New York, USA
`arthur.jacot@nyu.edu`

## ABSTRACT

When a deep ReLU network is initialized with small weights, gradient descent (GD) is at first dominated by the saddle at the origin in parameter space. We study the so-called escape directions along which GD leaves the origin, which play a similar role as the eigenvectors of the Hessian for strict saddles. We show that the optimal escape direction features a *low-rank bias* in its deeper layers: the first singular value of the $\ell$-th layer weight matrix is at least $\ell^{\frac{1}{4}}$ larger than any other singular value. We also prove a number of related results about these escape directions. We suggest that deep ReLU networks exhibit saddle-to-saddle dynamics, with GD visiting a sequence of saddles with increasing bottleneck rank (Jacot, 2023a).

## 1 INTRODUCTION

In spite of the groundbreaking success of Deep Neural Networks (DNNs), the training dynamics of GD in these models remain ill-understood, especially when the number of hidden layers is large. A significant step in our understanding is the (relatively recent) realization that there exist multiple regimes of training in large neural networks: a kernel or lazy regime, where DNNs simply implement kernel methods w.r.t. the Neural Tangent Kernel (NTK) (Jacot et al., 2018; Du et al., 2019; Allen-Zhu et al., 2019), and an active (or rich) regime characterized by the presence of feature learning (Chizat & Bach, 2018a; Rotskoff & Vanden-Eijnden, 2018; Chizat & Bach, 2018b) and some form of sparsity such as a low-rank bias (Li et al., 2020; Gunasekar et al., 2017; Arora et al., 2019a).

The kernel regime is significantly simpler than the active one because the dynamics can be linearized around the initialization (Jacot et al., 2018; Lee et al., 2019), and the loss is approximately quadratic/convex in the region traversed by GD (Jacot et al., 2020) (it also satisfies the PL inequality (Liu et al., 2020)). This makes it possible to prove strong convergence guarantees (Du et al., 2019; Allen-Zhu et al., 2019) and apply generalization bounds from the kernel methods literature almost directly (Arora et al., 2019b; Bordelon et al., 2020). Our understanding of the kernel regime is essentially complete, but some functions are unlearnable in the kernel regime yet learnable in the active regime (Bach, 2017; Ghorbani et al., 2020).

There are arguably many active regimes corresponding to different ways to leave the kernel regime, including small weight initialization (Woodworth et al., 2020), large learning rate (Lewkowycz et al., 2020; Damian et al., 2022), large noise in training (Smith et al., 2021; Pesme et al., 2021; Vivien et al., 2022; Wang & Jacot, 2024), late training with the cross-entropy loss (Ji & Telgarsky, 2018; Chizat & Bach, 2020), and weight decay (E et al., 2019; Ongie et al., 2020; Jacot, 2023a).

---

*Ioannis Bantzis was supported by a scholarship for graduate studies from the Onassis Foundation.

We will focus on the effect of initialization scale, where a phase change from kernel regime to active regime occurs as the variance of the initial weights decays towards zero. Here again we can distinguish two active regimes (Luo et al., 2021): the mean-field regime which lies right at transition between regimes (Chizat & Bach, 2018a; Rotskoff & Vanden-Eijnden, 2018; Mei et al., 2018), and the saddle-to-saddle regime (Saxe et al., 2014; Jacot et al., 2022; Pesme & Flammarion, 2023; Boursier et al., 2022) for even smaller initialization.

The mean-field limit was first described for shallow networks (Chizat & Bach, 2018a; Rotskoff & Vanden-Eijnden, 2018; Mei et al., 2018), and has more recently been extended to the deep case (Araújo et al., 2019; Bordelon & Pehlevan, 2022). A limitation of these approaches is that the limiting dynamics remain complex, especially in the deep case where they are described by algorithms that are not only very costly in the worst case (Bordelon & Pehlevan, 2022; Yang & Hu, 2020), but also difficult to interpret and reason about. This high complexity could be explained by the fact that the mean-field limit is critical, i.e. it lies exactly at the transition between kernel and active, and therefore it must capture the complexity of both of those regimes, as well as of the whole spectrum of intermediate dynamics.

## 1.1 SADDLE-TO-SADDLE DYNAMICS

This motivates the study of the saddle-to-saddle regime for even smaller initializations. As the name suggests, this regime is characterized by GD visiting a number of saddles before reaching a global minimizer. Roughly speaking, because of the small initializations, GD starts in the vicinity of the saddle which lies at the origin in parameter space and remains stuck there for a number of steps until it finds an escape direction, leading to a sudden drop in the loss. This escape direction exhibits a form of approximate sparsity (amongst other properties) that is preserved by GD. At this point, the level of sparsity can either be enough to fit the training data in which case the loss will drop to zero and training will stop, but if the network is 'too sparse' to fit the data, GD will approach another saddle at a lower cost (which is locally optimal given the sparsity) before escaping along a less sparse escape direction. GD can visit a sequence of saddles before reaching a final network that fits the data while being as sparse as possible. This has been described as performing a greedy algorithm (Li et al., 2020) where one tries to find the best data-fit with a sparsity constraint that is gradually weakened until the training data can be fitted.

Such incremental learning dynamics were first observed in diagonal linear networks (Saxe et al., 2014; 2019; Gidel et al., 2019) (and by extension to linear Convolutional Neural Networks, which are diagonal nets in Fourier space), before being extended to linear fully-connected networks (Arora et al., 2019a; Li et al., 2020; Jacot et al., 2022; Tu et al., 2024; Kunin et al., 2024). These result in coordinate sparsity of the learned vector for diagonal networks and rank sparsity of the learned matrix for fully-connected networks.

For nonlinear networks, the focus has been mainly on shallow networks, where a condensation effect is observed, wherein groups of neurons end up having the same activations (up to scaling). Roughly speaking, in the first escape direction, a first group of hidden neurons comes out first, all with the same activation (up to scaling), in the subsequent saddles, new groups can emerge or an existing group can split in two (Chizat & Bach, 2018a) (though sometimes they may fail to split leading to problems (Boursier & Flammarion, 2024)). This condensation effect leads to a form of sparsity, since each group then behaves as a single neuron, thus reducing the effective number of neurons (Luo et al., 2021; Simsek et al., 2021). These dynamics could be understood as implicitly implementing a Frank-Wolfe algorithm (Bach, 2017): alternating between finding new neurons to 'add' to the mix, and then tuning the mixing weights to get the best possible fit (Kunin et al., 2025).

To our knowledge, all prior theoretical analysis of saddle-to-saddle dynamics in deep nonlinear networks rely on an equivalence to deep linear networks, which can arises with differentiable nonlinearities (e.g. arctan) allowing for a Taylor approximation of the origin (Bai et al., 2022), or in specific settings where the ReLUs do not change signs (Zhang et al., 2025). This leads to a low-rank bias, where all layers are rank-one in the first escape direction. Saddle-to-saddle dynamics with multiple plateaus/saddles have been observed empirically in deep ReLU networks trained on supervised (Atanasov et al., 2024) and self-supervised (Simon et al., 2023) tasks, and these empirics motivate our present theoretical study.

## 1.2 BOTTLENECK RANK INCREMENTAL LEARNING

Surprisingly, we show a more complex rank sparsity structure in deep ReLU networks: the majority of layers are rank-one (or approximately so), with possibly a few high-rank layers at the beginning of the network, in contrast to linear nets, shallow ReLU networks, and deep nets with differentiable nonlinearity where all layers are rank-one in the first escape.

This fits into the bottleneck structure and related bottleneck rank (BN-rank) observed in large depth ReLU networks trained with weight decay (Jacot, 2023a;b; Wen & Jacot, 2024; Jacot & Kaiser, 2024), where almost all layers share the same low rank, with a few higher rank layers located close to the input and output layers. Additionally, in the middle low-rank layers ("inside the bottleneck"), the preactivations are approximately non-negative, so that the ReLU approximates the identity.

The bottleneck rank $\mathrm{Rank}_{BN}(f)$ is a notion of rank for finite piecewise linear functions $f$, defined as the minimal integer $k^*$ such that $f$ can be decomposed $f = h \circ g$ with intermediate dimension $k^*$ (Jacot, 2023a). For large depths, it is optimal in the sense of minimizing the parameter norm to represent $f$ with a bottleneck structure, where the first few high-dim. layers represent $g$, followed by many rank $k^*$ layers representing the identity on the dimension $k^*$ intermediate representation, before using the last few layers to represent $h$.

Our results imply that the first escape direction of deep ReLU networks has BN-rank one, because almost all layers are approximately rank-one except a few high rank layers in the beginning. This is a "half" bottleneck structure, since it lacks high dimensional layers before the outputs, but it still fits within the BN-rank theory, suggesting that the BN-rank is the correct notion of sparsity in deep ReLU networks (rather than the traditional notion of rank).

We conjecture that deep ReLU networks exhibit similar saddle-to-saddle dynamics as linear networks, with the distinction that it is the BN-rank that gradually increases rather than the traditional rank.

## 1.3 CONTRIBUTIONS

In this paper, we give a description of the saddle at the origin in deep ReLU networks, and the possible escape directions that GD could take as it escapes this first saddle. As in (Jacot et al., 2022), each escape direction can be assigned an escape speed, and we show that the optimal escape speed is non-decreasing in depth (Proposition 3.2).

We then prove in Theorem 3.1 that the optimal escape directions feature a low-rank bias that gets stronger in deeper layers (i.e. layers closer to the output layer). More precisely the weight matrix $W_\ell$ and activations $Z_\ell^\sigma$ over the training set for $\ell = 1, \ldots, L$ are $\ell^{-\frac{1}{4}}$-approximately rank 1 in the sense that their second singular value is $O(\ell^{-\frac{1}{4}})$ times smaller than the first. Furthermore, deeper layers are also more linear, i.e. the effect of the ReLU becomes weaker.

Finally, we provide in Section 4 an example of a simple dataset whose optimal escape direction has the following structure: the first weight matrix is rank two, followed by rank-one matrices. This shows that the structure of our first result where the first layers are not approximately rank-one but the deeper layers are is not an artifact of our proof technique and reflects real examples. This contrasts with previous saddle-to-saddle dynamics, where all layers are approximately rank-one in the first escape direction.

## 2 SADDLE AT THE ORIGIN

We represent the training dataset $x_1, \ldots, x_N \in \mathbb{R}^{d_{in}}$ as a $d_{in} \times N$ matrix $X$. We consider a fully-connected neural network of depth $L$ with widths $n_0 = d_{in}, n_1, \ldots, n_L = d_{out}$ and ReLU nonlinearity $\sigma(x) = \max\{x, 0\}$. The $n_\ell \times N$ dimensional matrices of activation $Z_\ell^\sigma$ and preactivation $Z_\ell$ at the $\ell$-th layer are then defined recursively as

$$Z_0^\sigma = X$$
$$Z_\ell = W_\ell Z_{\ell-1}^\sigma$$
$$Z_\ell^\sigma = \sigma(Z_\ell),$$

for the $n_\ell \times n_{\ell-1}$ weight matrix $W_\ell$, $\ell = 1, \ldots, L$. The weight matrices $W_1, \ldots, W_L$ are the parameters of the network, and we concatenate them into a single vector of parameters $\theta$ of dimension $P = \sum_\ell n_\ell n_{\ell-1}$. The outputs of the network are the last layer's preactivations $Y_\theta = Z_L$.

We consider a general cost $C : \mathbb{R}^{d_{out} \times N} \to \mathbb{R}$ that takes the network outputs $Y_\theta$ and returns the loss $\mathcal{L}(\theta) = C(Y_\theta)$. The parameters $\theta(t)$ are then trained with gradient flow (GF) on the loss $\mathcal{L}$

$$\partial_t \theta(t) = -\nabla \mathcal{L}(\theta(t))$$

starting from a random initialization $\theta_0 \sim \mathcal{N}(0, \sigma_0^2)$ for a small $\sigma_0$.

One can easily check that the origin $\theta = 0$ is a critical point of the loss. Our analysis will focus on the neighborhood of this saddle, and for such small parameters the outputs $Y_\theta$ will be small, we can therefore approximate the loss as

$$\mathcal{L}(\theta) = C(Y_\theta) = C(0) + \text{Tr}\left[\nabla C(0)^T Y_\theta\right] + O(\|Y_\theta\|_F^2), \tag{1}$$

where $\nabla C(0)$ is an $n_L \times N$ matrix. Since we only care about the dynamics of gradient flow, the first term can be dropped. We will therefore mainly focus on the *localized loss* $\mathcal{L}_0(\theta) = \text{Tr}[G^T Y_\theta]$, writing $G = \nabla C(0)$ for simplicity.

The localized loss $\mathcal{L}_0$ can be thought of as resulting from zooming into origin. It captures the loss in the neighborhood of the origin. Note that since the ReLU is not differentiable, neither is the loss at the origin, so that we cannot use the traditional strategy of approximating $\mathcal{L}_0$ by a polynomial. However, this loss has the advantage of being homogeneous with degree $L$, i.e. $\mathcal{L}_0(\lambda\theta) = \lambda^L \mathcal{L}_0(\theta)$, which will be key in our analysis. A derivation of equation 1 and of this homogeneity can be found in Section A.1 of the Appendix.

## 2.1 Gradient Flow on Homogeneous Losses

On a homogeneous loss, the GF dynamics decompose into dynamics of the norm $\|\theta\|$ and of the normalized parameters $\bar{\theta} = \theta/\|\theta\|$ (see Proposition A.1 for a detailed derivation):

$$\partial_t \|\theta(t)\| = -\bar{\theta}(t)^T \nabla \mathcal{L}_0(\theta(t)) = -L\|\theta(t)\|^{L-1} \mathcal{L}_0(\bar{\theta}(t))$$
$$\partial_t \bar{\theta}(t) = -\|\theta(t)\|^{L-2}\left(I - \bar{\theta}(t)\bar{\theta}(t)^T\right)\nabla \mathcal{L}_0(\bar{\theta}(t)).$$

where we used Euler's homogeneous function theorem: $\theta^T \nabla \mathcal{L}_0(\theta) = L\mathcal{L}_0(\theta)$. In our setting the parameter norm is very small $\|\theta\| \ll 1$, which implies that the dynamics of $\bar{\theta}$ will be much faster than that of the norm $\|\theta\|$ because $\|\theta(t)\|^{L-2} \gg \|\theta(t)\|^{L-1}$.

Notice that $(I - \bar{\theta}\bar{\theta}^T)$ is the projection to the tangent space of the sphere at $\bar{\theta}$, which implies that the normalized parameters are doing projected GF over the unit sphere on the $\mathcal{L}_0$ loss (up to a prefactor of $\|\theta\|^{L-2}$ which can be interpreted as a speed up of the dynamics for larger norms).

Therefore, we may reparametrize time $t(q)$, such that $q(t) = \int_0^q \|\theta(q_1)\|^{L-2} dq_1$, which correspond to switching to a time-dependent learning rate $\eta_q = \|\theta(q)\|^{2-L}$, we obtain the dynamics:

$$\partial_q \|\theta(q)\| = -L\|\theta(q)\| \mathcal{L}_0(\bar{\theta}(q))$$
$$\partial_q \bar{\theta}(q) = -\left(I - \bar{\theta}(q)\bar{\theta}(q)^T\right)\nabla \mathcal{L}_0(\bar{\theta}(q)).$$

We can therefore solve for $\bar{\theta}(q)$ on its own, and the norm $\|\theta(q)\|$ then takes the form

$$\|\theta(q)\| = \|\theta(0)\| \exp\left(-L \int_0^q \mathcal{L}_0(\bar{\theta}(q_1)) dq_1\right).$$

If needed, these solutions can then be translated back in $t$-time, using the formula

$$t(q) = \int_0^q \|\theta(q_1)\|^{2-L} dq_1 = \|\theta(0)\|^{2-L} \int_0^q \exp\left(L(L-2)\int_0^{q_1} \mathcal{L}_0(\bar{\theta}(q_2)) dq_2\right) dq_1.$$

## 2.2 Escape Directions and their Speeds

Assuming convergence of the projected gradient flow $\bar{\theta}(q)$, for all initializations $x_0$ there will be a time $q_1$ where $\bar{\theta}(q_1)$ will be close to a critical point of $\mathcal{L}_0$ restricted to the sphere, i.e. a point $\bar{\theta}^*$ such that $\left(I - \bar{\theta}^* \bar{\theta}^{*T}\right)\nabla \mathcal{L}_0(\bar{\theta}^*) = 0$. We call these escape directions (assuming $\mathcal{L}_0(\bar{\theta}^*) < 0$), because once it is reached, $\bar{\theta}(q)$ remains approximately constant while the parameter norm grows fast.

**Definition 2.1.** *An **escape direction** is a vector on the sphere $\rho \in L^{1/2}\mathbb{S}^{P-1}$ such that $\nabla \mathcal{L}_0(\rho) = -s\rho$ for some $s \in \mathbb{R}_+$, which we call the **escape speed** associated with $\rho$. We switch from the unit sphere to the radius $\sqrt{L}$ sphere as it will lead to cleaner formulas.*

*An **optimal escape direction** $\rho^* \in L^{1/2}\mathbb{S}^{P-1}$ is an escape direction with the largest speed $s^* > 0$. It is a minimizer of $\mathcal{L}_0$ restricted to $L^{1/2}\mathbb{S}^{P-1}$:*

$$\rho^* \in \arg\min_{\rho \in L^{1/2}\mathbb{S}^{P-1}} \mathcal{L}_0(\rho).$$

If the parameters start aligned with an escape direction $\theta_0 \propto \rho$, then GF on the localized loss will diverge towards infinity in a straight line with rate determined by the depth $L$ and the escape speed $s$:

**Proposition 2.2.** *Considering gradient flow on the localized loss $\mathcal{L}_0$, if at some time $t_0$ the parameter satisfies*

$$\theta(t_0) = \rho \quad with \quad \rho \in L^{1/2}\mathbb{S}^{P-1} \quad and \quad \nabla \mathcal{L}_0(\rho) = -s\,\rho,$$

*then for all $t \geq t_0$ the normalized direction remains constant, and the norm $\|\theta(t)\|$ satisfies*

$$\|\theta(t)\| = \begin{cases} \left(\|\theta(t_0)\|^{2-L} + (2-L)L\,s\,(t-t_0)\right)^{\frac{1}{2-L}}, & if\ L \neq 2, \\ \|\theta(t_0)\|\,\exp\left(2\,s\,(t-t_0)\right), & if\ L = 2. \end{cases}$$

Of course, the localized loss $\mathcal{L}_0$ is only a good approximation as long as the outputs $Y_\theta$ are small. This will apply up to some escape time $t_1(r)$ which is the first time the outputs satisfy $\|Y_\theta\|_F = r$. Since $\|Y_{\theta(t)}\|_F \leq \|W_L\|_{op} \cdots \|W_1\|_{op}\|X\|_F \leq \left(\frac{\|\theta\|}{\sqrt{L}}\right)^L \|X\|_F$ we know that

$$t_1(r) - t_0 \geq \begin{cases} \frac{1}{(L-2)Ls}\left[\|\theta(t_0)\|^{2-L} - L^{\frac{2-L}{2}} r^{\frac{2-L}{L}}\right], & if\ L \neq 2, \\ \frac{1}{2s}(\log \sqrt{L} r^{\frac{1}{L}} - \log\|\theta(t_0)\|), & if\ L = 2. \end{cases}$$

After this escape time, we expect the localized GF to diverge from the true GF: the localized GF diverges towards infinity (in finite time when $L > 2$), while the true GF typically slows down as it approaches another saddle or a minima. This paper focuses on the dynamics before the escape time.

In general, we do not start aligned with an escape direction, but since the normalized parameters $\bar{\theta}(q)$ follow GF restricted to the sphere , they will converge to an escape direction, at which point a similar explosion of the norm will take place.

Note that the dynamics of $\bar{\theta}(q)$ (in reparametrized $q$-time) are unaffected by multiplying the initialization $\theta_0$ by a factor $\alpha > 0$. Therefore the time $q_1$ of convergence to an escape direction is independent of $\alpha$, and at the time $q_1$, the parameter norm will depend linearly on $\alpha$: $\|\theta(q_1)\| = C\alpha$ for some $C > 0$. We can therefore always choose a small enough $\alpha$ so that the Taylor approximation (Equation 1) remains valid up to the time of convergence $q_1$.

The convergence of GF to escape directions is essentially the same as the convergence to Karush-Kuhn-Tucker (KKT) points as GD diverges towards infinity with the cross-entropy loss Ji & Telgarsky (2018); Chizat & Bach (2020). The techniques used in these papers could be used to extend the present GF analysis to a GD convergence.

## 3 Low Rank Bias and Approx. Linearity of the Escape Directions

The main result of this paper is that at the optimal escape directions, the deeper layers (i.e. for large $\ell$) are approximately low-rank and have almost no nonlinearity effect:

**Theorem 3.1.** *Consider an optimal escape direction*

$$\theta^\star = \arg\min_{\|\theta\|^2=L} \mathrm{Tr}\big[G^\top Y_\theta\big]$$

*with optimal speed* $s^* = min_{\|\theta\|^2=L} \mathrm{Tr}\big[G^\top Y_\theta\big]$, *then for all layers* $\ell$ *with* $\ell > c^2$ *we have:*

$$\frac{\sum_{i\geq 2} s_i^2(W_\ell)}{\sum_{i\geq 1} s_i^2(W_\ell)}, \frac{\sum_{i\geq 2} s_i^2(Z_\ell^\sigma)}{\sum_{i\geq 1} s_i^2(Z_\ell^\sigma)}, \frac{\|Z_\ell^\sigma - Z_\ell\|_F^2}{\|Z_\ell\|_F^2} \leq 8\frac{c}{1-c\ell^{-\frac{1}{2}}}\ell^{-\frac{1}{2}}$$

*where* $s_i(A)$ *is the* $i$-*th largest singular value of* $A$ *and* $c = \frac{\|X\|_F\|G\|_F}{s^*}\sqrt{2\log\frac{\|X\|_F\|G\|_F}{s^*}}$.

In the rest of the section, we will prove a result that shows that the optimal escape speed $s^*$ is increasing in depth, thus controlling the constant $c$ in depth. This guarantees that the condition $\ell > c^2$ holds for all but finitely many of the initial layers of the network. We then present a sketch of proof for the Theorem, stating a few intermediate results that are of independent interest.

## 3.1 OPTIMAL SPEED IS INCREASING IN DEPTH

The bounds of Theorem 3.1 are strongest when the optimal escape direction $s^*$ is large. Thankfully, the optimal escape speed is increasing in $L$:

**Proposition 3.2.** *Given a depth $L$ network with $\mathcal{L}_0(\theta) = -s_0$ for $s_0 > 0$ and $\|\theta\|^2 = L$, we can construct a network of depth $L + k$ for any $k \geq 1$ with parameters $\theta'$ that satisfies $\|\theta'\|^2 = L + k$ and $\mathcal{L}_0(\theta') \leq \mathcal{L}_0(\theta)$. Therefore, the optimal escape speed $s^*(L)$ is a non-decreasing function.*

*Furthermore, in the deeper network, we have* $\mathrm{Rank}(Z_{L'}) = \mathrm{Rank}\, W_{L'} = 1$ *for all* $L' \geq L$ *and* $Z_{L'} = Z_{L'}^\sigma$ *for all* $L' > L$.

To construct the deeper network, we first transform the last weights $W_L$ to be rank-one (this is possible without increasing $\mathcal{L}_0$), and we then add rank-one weights in the additional layers. Some very similar structures have been used in previous work (Jacot, 2023a; Bai et al., 2022).

## 3.2 SKETCH OF PROOF

To prove Theorem 3.1, we first show that if the inputs are approximately rank-one, then the optimal escape will be approximately rank-one in all layers:

**Proposition 3.3.** *Consider the minimizer $\theta^* = \arg min_{\|\theta\|^2\leq L} \mathrm{Tr}\big[G^\top Y_\theta(uv^\top + X)\big]$ where $u, v \in \mathbb{R}^n$, $u, v \geq 0$ entry wise and $\|X\|_F \leq \epsilon$ for some $\epsilon > 0$. Then for all $\ell$ we have:*

$$\frac{\sum_{i\geq 2} s_i^2(W_\ell)}{\sum_{i\geq 1} s_i^2(W_\ell)}, \frac{\sum_{i\geq 2} s_i^2(Z_\ell^\sigma)}{\sum_{i\geq 1} s_i^2(Z_\ell^\sigma)}, \frac{\|Z_\ell^\sigma - Z_\ell\|_F^2}{\|Z_\ell\|_F^2} \leq 8\frac{\|G\|_F}{s^* - \|G\|_F\epsilon}\epsilon.$$

This also implies that if a hidden representation is approximately rank-one in one layer $\ell_0$, then it must also be approximately rank-one in all subsequent layers $\ell \geq \ell_0$. We can prove the existence of many such low-rank layers assuming the escape speed is large enough:

**Proposition 3.4.** *Assuming $Tr[G^\top Z_L] \leq -s_0$ for some constant $s_0 > 0$ and $\|\theta\|^2 \leq L$ then for any ratio $p \in (0, 1)$ there are at least $(1 - p)L$ layers that are approximately rank-one in the sense that*

$$\frac{\sum_{i\geq 2} s_i^2(Z_\ell^\sigma)}{\sum_{i\geq 1} s_i^2(Z_\ell^\sigma)} \leq 2\log\left(\frac{\|X\|_F\|G\|_F}{s_0}\right)\frac{1}{pL}$$

The proof of Theorem 3.1 therefore goes as follows: for any $\ell = pL$, Proposition 3.4 implies that there are at least $(1 - p)L = L - \ell$ layers that are approximately rank-one. The earliest such layer $\ell_0$ must satisfy $\ell_0 \leq \ell$. Proposition 3.3 implies that all layers after $\ell_0$ must be approximately rank-one, including the $\ell$-th layer.

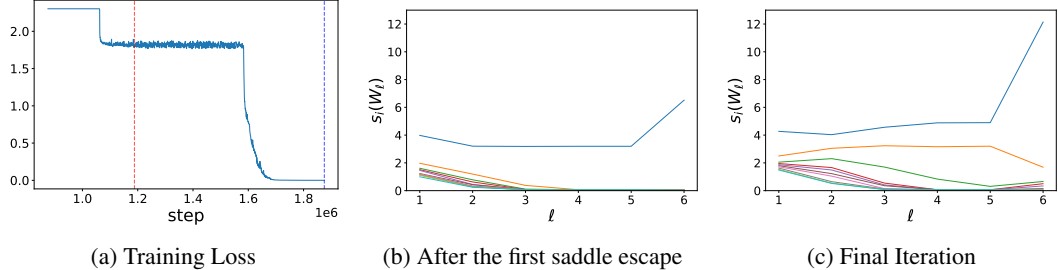

| (a) Training Loss | (b) After the first saddle escape | (c) Final Iteration |

Figure 1: **Deeper layers show a stronger bias toward low-rank structure than earlier layers on MNIST. Left:** Training loss over training time. Vertical lines indicate the specific iterations at which singular values are extracted. **Center and Right:** Top 10 singular values of the weight matrices per layer $\ell$ for layers 1–6 including input and output layer.

The two propositions are also of independent interest. Proposition 3.3 gives an example of inputs where all layers are low-rank, not just the deeper layers. Proposition 3.4 applies to any parameter with fast enough escape speed, not just to the optimal escape direction, and guarantees a similar low rank structure. Interestingly, in contrast to the other results, it does not say anything about where those low-rank layers are.

### 3.3 EMPIRICAL RESULTS ON MNIST

We empirically confirm the presence of low-rank structure in networks trained on the MNIST dataset. Specifically, we train a 6-layer fully connected network without bias and with small initialization.

Figure 1 highlights two distinct saddle points during training. After escaping the first saddle, we observe the emergence of a single dominant singular value in every layer, with this effect being particularly pronounced in the deeper layers (layers 4–6). While our theoretical analysis explains the behavior after the first saddle escape, our experiments reveal that, towards the end of training, a second dominant singular value appears. This suggests that the rank of the weight matrices increases following subsequent saddle escapes. A detailed visualization of the singular value evolution in each layer is provided in Figure 3 in the Appendix.

## 4 THE OPTIMAL ESCAPE DIRECTION IS NOT ALWAYS EXACTLY RANK ONE

Our discussion has thus far consisted of results which paint the picture that *deep ReLU networks trained from small initialization first escape the origin in a direction which is approximately rank one in each weight matrix.* Much of our labor has been in identifying suitable notions of "approximately rank one." Before concluding, it is worth asking: *do we actually need such notions?* In fact, if one performs straightforward numerical experiments on simple datasets, one will often find that the first escape direction is *exactly* rank one in each layer. Might we hope that the optimal escape direction is in fact *always exactly rank one?*

In this section, we provide a simple counterexample in which the optimal escape direction is rank *two* in the first layer. We then give numerical experiments which show that (projected) gradient descent actually finds this rank-two solution.

**Example 1** (Rank-two optimal escape direction). *Consider the unit circle dataset $(x_j)_{j=1}^N = \left(\sin\left(\frac{2\pi * j}{N}\right), \cos\left(\frac{2\pi * j}{N}\right)\right)_{j=1}^N$ with alternating loss gradients $G = ((-1)^j)_{j=1}^N$.[1] Let $N = 8$. Consider training a depth-three bias-free ReLU MLP with hidden width at least four from small initialization on this dataset. Then the optimal rank-one escape direction has speed $s_1 = \sqrt{2} - 1 \approx 0.414$, but there exists a better rank-two escape direction with speed $s_2 = \frac{1}{2}$.*

---

[1]Such alternating loss gradients can result straightforwardly from, for example, targets $Y = ((-1)^j)_{j=1}^N$ and the usual squared loss.

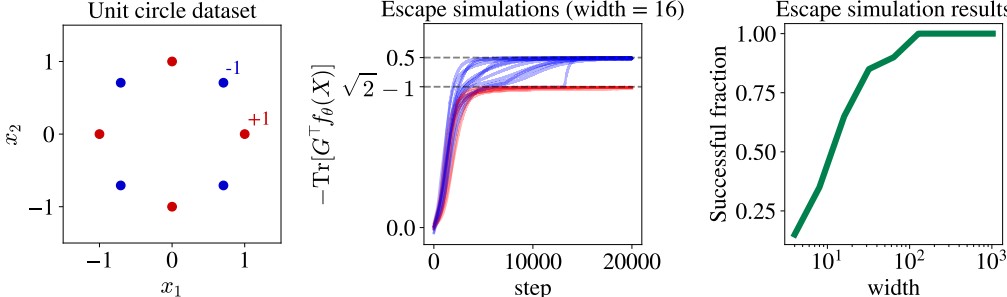

Figure 2: **Depth-3 neural networks find rank-two escape directions on a toy dataset. Left:** visualization of the dataset. Red and blue points have loss gradient values $G = 1$ and $G = -1$, respectively. **Center:** several training runs of projected gradient descent on the first-order loss objective under the parameter norm constraint $\|\theta\|^2 = L$. Runs whose objective exceeds $\sqrt{2} - 1$, the best achievable value for rank-one weights, are colored blue and deemed successful. **Right:** as width increases, the fraction of successful runs increases. See Figure 8 for a visualization of the training runs at all widths.

**Proof.** Our network has weight matrices $W_1, W_2, W_3$ which parameterize the network function as $Y_\theta = W_3\sigma \circ W_2\sigma \circ W_1 X$. As discussed in Subsection 2.2, we wish to minimize the escape speed $s = -Tr[G^\top Y_\theta]$ such that $\sum_\ell \|W_\ell\|_F^2 = 3$. We know from homogeneity that the minimizer will have $\|W_\ell\|_F = 1$ for all $\ell$.

If we additionally constrain all three weight matrices to be rank-one, then a width-one ReLU network can achieve the same maximal escape speed (a network with only rank 1 layers can only represent 'one neuron functions': $Y_\theta = u\sigma(v^\top x)$ for some vectors $u, v$, independently of depth). Taking into account the positivity of $\mathrm{ReLU}$, we need only study a width-one network with $W_1 = [\cos(\phi), \sin(\phi)]$ for some $\phi \in [0, 2\pi)$, $W_2 = [1]$, and $W_3 = [\pm 1]$. The only degree of freedom remaining is the angle $\phi$ to which $W_1$ is attuned. We solve this 1D optimization problem in Appendix D.1, finding that the optima fall at $\phi = \frac{\pi j}{4}$ for $j \in \mathbb{Z}$, giving speed $s_1 = \sqrt{2} - 1 \approx 0.414$.

Without such a rank-one constraint, we can improve this speed. We use only four neurons in the first hidden layer and one neuron in the second hidden layer (setting all incoming and outgoing weights to other neurons to zero) and choose the following weights for the active neurons:

$$W_1 = \tfrac{1}{2}\begin{bmatrix} 1 & 0 \\ 0 & 1 \\ -1 & 0 \\ 0 & -1 \end{bmatrix}, \quad W_2 = \tfrac{1}{2}\begin{bmatrix} 1 & -1 & 1 & -1 \end{bmatrix}, \quad W_3 = [1]. \tag{2}$$

This gives a speed $s_2 = \frac{1}{2}$.

This counterexample shows that the optimal escape direction may in fact be non-rank-one, and thus it is reasonable to search for a sense in which, for a sufficiently deep network, the optimal escape direction is *approximately* rank one.[2]

### 4.1 NUMERICAL EXPERIMENTS: WIDE NETWORKS FIND THE OPTIMAL ESCAPE DIRECTION

Of course, the existence of such a non-rank-one optimal escape direction is only interesting if gradient descent actually finds it. In this case, it does. We train networks of varying width with projected gradient descent to minimize the loss on the sphere $\|\theta\|^2 = 3$. As shown in Figure 2, wider networks are more likely to converge to the faster, rank-two escape direction.

---

[2]It is worth noting that there may exist an even faster escape direction than the rank-two solution we identify (though we doubt it; see numerical experiments), but in any case we may be assured that the fastest escape direction is *not* rank one.

## 5 DISCUSSION: SADDLE-TO-SADDLE DYNAMICS

The results of this paper only describe the very first step of a much more complex training path. They describe the escape from the first saddle at origin, but it is likely that the full dynamics might visit the neighborhood of multiple saddles, as is the case for linear networks (Jacot et al., 2022; Li et al., 2020) or shallow ReLU networks (Abbe et al., 2021; 2022). We now state a few conjectures/hypotheses, which should be viewed as possible next steps towards the goal of describing the complete Saddle-to-Saddle dynamics:

**(1) Large width GD finds the optimal escape direction:** Our numerical experiments suggest that wider networks are able to find the optimal escape direction with GD, even when this optimal escape direction has some higher rank layers. The intuition is that the more neurons, the more likely it is that a subset of neurons implement a 'circuit' that is similar to an optimal escape direction, and that this group will out-compete the other neurons and end up dominating. Note that even in shallow networks, finding this optimal escape direction is known to be NP-hard (Bach, 2017), which implies that an exponential number of neurons might be required in the worst case.

**(2) Exact rank-one at most layers:** Inspired by our illustrating example, we believe that it is likely that the optimal escape directions might only have a finite number of high-rank layers at the beginning, followed by rank-one identity layers until the outputs.

Note that if we assume that the optimal speed direction $s^*(L)$, plateaus after a certain $L_0$, i.e. $s^*(L) = s^*(L_0)$ for all $L \geq L_0$, then Proposition 3.2 already implies that there is an optimal escape direction where all layers $\ell \geq L_0$ are rank 1. Conversely, if there is an optimal escape directions with only rank-one layers after the $L_0$-th layer, then $s^*(L) = s^*(L_0)$ for all $L \geq L_0$.

**(3) rank-one layers remain rank-one until the next saddle:**

Assuming that GD does find the optimal escape direction, it will have approximately rank-one layers as it escapes the saddle. The next step is to show that these layers remain approximately rank-one until reaching a second saddle.

In linear networks, this follows from the fact that the optimal escape direction is rank-one and balanced (i.e. $W_\ell^T W_\ell = W_{\ell-1} W_{\ell-1}^T$ for all layers $\ell$), and that the space of rank-one and balanced network is an invariant space under GF.

The ReLU case is more difficult because we have only approximately rank-one layers. More precisely to guarantee that there is a layer that is $\epsilon$-approximately rank-one, we need both a small initialization and a large depth, in contrast to linear networks where a small enough initialization is sufficient. Our second conjecture would help with this aspect.

The next difficulty is to show that the approximate rank-one layers remain so for a sufficient amount of time. The key tool to prove this in linear networks is balancedness. ReLU networks only satisfy weak balancedness in general , i.e. $diag(W_\ell^T W_\ell) = diag(W_{\ell-1} W_{\ell-1}^T)$, however the stronger balancedness applies at layers where the pre-activations have non-negative entries: $Z_\ell \geq 0$.

**(4) BN-rank incremental learning** The final goal is to prove that these Saddle-to-Saddle dynamics allow ReLU networks to implement a form of greedy low BN-rank algorithm, which searches first among BN-rank one functions, then among gradually higher rank functions, stopping at the smallest BN-rank sufficient to fit the data.

Again, this is inspired by an analogy to linear network, which implement a greedy low-rank algorithm to minimize the traditional rank. In parameter space, the GD dynamics visits a sequence of saddles of increasing rank. It starts close to the saddle at the origin (the best rank 0 fit), before escaping along a rank-one direction until reaching a rank-one critical point (locally optimal rank-one fit). If the loss is zero at this point, the GD dynamics stop, otherwise this best rank-one fit is a saddle where GD plateaus for some time until escaping along a rank 2 direction, and so on (Jacot et al., 2022).

The so-called Bottleneck rank $\mathrm{Rank}_{BN}(f)$ (Jacot, 2023a) is the smallest integer $k$ such that $f$ can be represented as the composition of two functions $f = h \circ g$ with inner dimension $k$. Several recent papers have shown how the BN rank plays a central role in deep ReLU networks trained with weight-decay/$L_2$-regularization (Jacot, 2023a;b; Wen & Jacot, 2024; Jacot & Kaiser, 2024). In particular, these works observe the emergence of a bottleneck structure as the depth grows, where all middle layers of the network share the same low rank (discarding small singular values of $W_\ell$), which

equals the BN rank of the network, with only a finite number of high-rank layers at the beginning and end of the network.

Our results can be interpreted as saying that the optimal escape direction of the saddle at the origin exhibits a 'half-bottleneck' (because there are only high-dimensional layers at the beginning of the network, not at the end) with BN-rank one. This suggests that the Saddle-to-Saddle dynamics in deep ReLU networks could correspond to a greedy low-BN-rank search, where the BN-rank increases gradually between each plateau/saddle. Interestingly, previous theoretical analysis of the bottleneck structure were only able to prove the existence of low-rank layers but not necessarily locate them (Jacot, 2023b), our ability to prove that the deeper layers are all approximately low-rank is therefore a significant improvement over the previous proof techniques.

It is possible that in contrast to linear network, the complete Saddle-to-Saddle dynamics would require both a small initialization and large depth. This matches our numerical experiments in Figure 1 and Figure 4 in the appendix, where we observe more distinct plateaus in depth 6 layer compared to a depth 4 layer. This suggests that in contrast to linear networks, where the plateaus can be made longer and more distinct by taking smaller initialization, for ReLU networks we need to also increase the depth to achieve the same effect.

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

## A    PROOFS OF THEOREMS

### A.1    SMALL DERIVATIONS

Let us first a few derivations of equations stated in the main.

First, Equation 1, which follows from a Taylor expansion of the cost $C$ w.r.t. the output labels:

$$\mathcal{L}(\theta) = C(Y_\theta)$$
$$= C(0) + \sum_{ij} \partial_{Y_{ij}} C(Y_\theta) Y_{\theta,ij} + O\left(\|Y_\theta\|_F^2\right)$$
$$= C(0) + \text{Tr}\left[\nabla C(0)^T Y_\theta\right] + O(\|Y_\theta\|_F^2).$$

As a concrete example, we have for the Mean Squared Error $C(Y) = \frac{1}{N}\|Y - Y^*\|_F^2$ for some 'true' labels $Y^*$, the equivalent three terms are:

$$\mathcal{L}(\theta) = \frac{1}{N}\|Y - Y^*\|_F^2 = \frac{1}{N}\|Y^*\|_F^2 - \frac{2}{N}\text{Tr}[Y^{*T} Y_\theta] + \frac{1}{N}\|Y_\theta\|_F^2,$$

since $\nabla C(0) = -\frac{2}{N} Y^*$.

Second, the homogeneity of localized loss $\mathcal{L}_0(\theta) = \text{Tr}[\nabla C(0)^T]$ follows from the fact that if all parameters are rescaled by a factor of $\lambda$ then the first activation will be multiplied by $\lambda$ (because $\alpha_1(x) = \sigma((\lambda W_1)x) = \lambda\sigma(W_1 x)$ ), the second activation by $\lambda^2$ (because $\alpha_2(x) = \sigma((\lambda W_2)(\lambda\alpha_1(x))) = \lambda^2\sigma(W_1\alpha_1(x)))$ and so on until the outputs which are multiplied by $\lambda^L$. The localized loss will have the same prefactor of $\lambda^L$ thanks to the linearity of the trace:

$$\mathcal{L}_0(\lambda\theta) = \text{Tr}[\nabla C(0)^T(\lambda^L Y_\theta)] = \lambda^L \mathcal{L}_0(\theta).$$

### A.2    GRADIENT FLOW ON HOMOGENEOUS LOSSES

**Proposition A.1.** *On a homogeneous loss, the GF dynamics decompose into dynamics of the norm $\|\theta\|$ and of the normalized parameters $\bar{\theta} = \theta/\|\theta\|$:*

$$\partial_t\|\theta(t)\| = -\bar{\theta}(t)^T\nabla\mathcal{L}_0(\theta(t)) = -L\|\theta(t)\|^{L-1}\mathcal{L}_0(\bar{\theta}(t))$$
$$\partial_t\bar{\theta}(t) = -\|\theta(t)\|^{L-2}\left(I - \bar{\theta}(t)\bar{\theta}(t)^T\right)\nabla\mathcal{L}_0(\bar{\theta}(t)).$$

*Proof.* Since $\mathcal{L}_0$ satisfies gradient flow with respect to $\theta$, we have

$$\frac{d\theta}{dt} = -\nabla\mathcal{L}_0(\theta).$$

Because $\mathcal{L}_0$ is $L$-homogeneous, Euler's homogeneous function theorem implies:

$$\theta^\top\nabla\mathcal{L}_0(\theta) = L\,\mathcal{L}_0(\theta).$$

Now, define the normalized parameter

$$\bar{\theta} = \frac{\theta}{\|\theta\|}.$$

Differentiating $\bar{\theta}$ with respect to time $t$ using the quotient rule yields:

$$\frac{d\bar{\theta}}{dt} = \frac{d}{dt}\left(\frac{\theta}{\|\theta\|}\right) = \frac{\frac{d\theta}{dt}\|\theta\| - \theta\frac{d\|\theta\|}{dt}}{\|\theta\|^2}.$$

Substituting $\frac{d\theta}{dt} = -\nabla\mathcal{L}_0(\theta)$ gives:

$$\frac{d\bar{\theta}}{dt} = \frac{-\nabla\mathcal{L}_0(\theta)\|\theta\| - \theta\frac{d\|\theta\|}{dt}}{\|\theta\|^2}.$$

To compute $\frac{d\|\theta\|}{dt}$, note that

$$\|\theta\| = (\theta^\top \theta)^{1/2}.$$

Differentiating, we obtain:

$$\frac{d\|\theta\|}{dt} = \frac{1}{\|\theta\|}\theta^\top \frac{d\theta}{dt} = \frac{1}{\|\theta\|}\theta^\top \big(-\nabla\mathcal{L}_0(\theta)\big).$$

Using the homogeneity property $\theta^\top \nabla\mathcal{L}_0(\theta) = L\,\mathcal{L}_0(\theta)$, this simplifies to:

$$\frac{d\|\theta\|}{dt} = -\frac{L\,\mathcal{L}_0(\theta)}{\|\theta\|}.$$

Substitute this back into the expression for $\frac{d\bar{\theta}}{dt}$:

$$\frac{d\bar{\theta}}{dt} = \frac{-\nabla\mathcal{L}_0(\theta)\,\|\theta\| + \theta\,\frac{L\,\mathcal{L}_0(\theta)}{\|\theta\|}}{\|\theta\|^2}.$$

This can be simplified as:

$$\frac{d\bar{\theta}}{dt} = \frac{-\nabla\mathcal{L}_0(\theta)}{\|\theta\|} + \frac{\theta}{\|\theta\|^3}\,L\,\mathcal{L}_0(\theta).$$

We wish to express the right-hand side in terms of $\bar{\theta}$. Using the scaling property of the gradient for a homogeneous function, we have

$$\nabla\mathcal{L}_0(\theta) = \|\theta\|^{L-1}\nabla\mathcal{L}_0(\bar{\theta}),$$

and recalling that

$$\theta^\top \nabla\mathcal{L}_0(\theta) = L\,\mathcal{L}_0(\theta),$$

we finally obtain:

$$\frac{d\bar{\theta}}{dt} = -\|\theta\|^{L-2}\nabla\mathcal{L}_0(\bar{\theta}) + \|\theta\|^{L-4}\theta\,\theta^\top \nabla\mathcal{L}_0(\bar{\theta}) = -\|\theta\|^{L-2}\Big(I - \bar{\theta}\bar{\theta}^\top\Big)\nabla\mathcal{L}_0(\bar{\theta}).$$

$\square$

## A.3 EXPLOSION IN ESCAPE DIRECTION

**Proposition A.2.** *If at some time $t_0$ the parameter satisfies*

$$\theta(t_0) = \rho \quad with \quad \rho \in L^{1/2}\mathbb{S}^{P-1} \quad and \quad \nabla\mathcal{L}_0(\rho) = -s\,\rho,$$

*then for all $t \geq t_0$ the normalized direction remains constant, and the norm $\|\theta(t)\|$ satisfies*

$$\|\theta(t)\| = \begin{cases} \big(\|\theta(t_0)\|^{\,2-L} + (2-L)L\,s\,(t-t_0)\big)^{\frac{1}{2-L}}, & if\ L \neq 2, \\ \|\theta(t_0)\|\,\exp\Big(2\,s\,(t-t_0)\Big), & if\ L = 2. \end{cases}$$

*Proof.* Using the chain rule we have

$$\frac{d}{dt}\|\theta(t)\| = \frac{1}{\|\theta(t)\|}\theta(t)^\top \frac{d\theta}{dt} = -\frac{1}{\|\theta(t)\|}\theta(t)^\top \nabla\mathcal{L}_0\big(\theta(t)\big).$$

Using Euler's theorem,

$$\theta(t)^\top \nabla\mathcal{L}_0\big(\theta(t)\big) = L\,\mathcal{L}_0\big(\theta(t)\big),$$

we obtain

$$\frac{d}{dt}\|\theta(t)\| = -\frac{L}{\|\theta(t)\|}\,\mathcal{L}_0\big(\theta(t)\big).$$

Since $\theta(t) = \|\theta(t)\|\bar{\theta}(t)$ and by homogeneity

$$\mathcal{L}_0\big(\theta(t)\big) = \|\theta(t)\|^L\,\mathcal{L}_0\big(\bar{\theta}(t)\big),$$

and because $\bar{\theta}(t) = \bar{\theta}(t_0)$ for all $t \geq t_0$ with $\mathcal{L}_0\big(\bar{\theta}(t_0)\big) = -s$, we deduce

$$\mathcal{L}_0\big(\theta(t)\big) = -s\,\|\theta(t)\|^L.$$

Substituting this back, we have

$$\frac{d}{dt}\|\theta(t)\| = -\frac{L}{\|\theta(t)\|}\left(-s\,\|\theta(t)\|^L\right) = L\,s\,\|\theta(t)\|^{L-1}.$$

Defining $R(t) = \|\theta(t)\|$, the above becomes the separable ordinary differential equation

$$\frac{dR}{dt} = L\,s\,R^{L-1}, \quad R(t_0) = \|\theta(t_0)\|.$$

Case 1: $L \neq 2$. We separate variables:

$$R^{1-L}\,dR = L\,s\,dt.$$

Integrate from $t_0$ to $t$:

$$\int_{R(t_0)}^{R(t)} R^{1-L}\,dR = \int_{t_0}^{t} L\,s\,d\tau.$$

The left-hand side integrates to

$$\left.\frac{R^{2-L}}{2-L}\right|_{R(t_0)}^{R(t)} = \frac{R(t)^{2-L} - R(t_0)^{2-L}}{2-L}.$$

Hence,

$$\frac{R(t)^{2-L} - R(t_0)^{2-L}}{2-L} = L\,s\,(t-t_0).$$

Solving for $R(t)$ gives

$$R(t)^{2-L} = R(t_0)^{2-L} + (2-L)L\,s\,(t-t_0),$$

or equivalently,

$$\|\theta(t)\| = \left(\|\theta(t_0)\|^{2-L} + (2-L)L\,s\,(t-t_0)\right)^{\frac{1}{2-L}}.$$

Case 2: $L = 2$. The ODE reduces to

$$\frac{dR}{dt} = 2\,s\,R,$$

which is linear. Its unique solution is

$$R(t) = R(t_0)\,\exp\big(2\,s\,(t-t_0)\big),$$

that is,

$$\|\theta(t)\| = \|\theta(t_0)\|\,\exp\big(2\,s\,(t-t_0)\big).$$

$\square$

## A.4 Optimal Speed is Increasing in Depth

**Proposition A.3.** *Given a depth $L$ network with $\mathcal{L}_0(\theta) = -s_0$ for $s_0 > 0$ and $\|\theta\|^2 = L$, we can construct a network of depth $L + k$ for any $k \geq 1$ with parameters $\theta'$ that satisfies $\|\theta'\|^2 = L + k$ and $\mathcal{L}_0(\theta') \leq \mathcal{L}_0(\theta)$. Therefore, the optimal escape speed $s^*(L)$ is a non-decreasing function.*

*Furthermore, in the deeper network, we have $\mathrm{Rank}(Z_{L'}) = \mathrm{Rank}\,W_{L'} = 1$ for all $L' \geq L$ and $Z_{L'} = Z_{L'}^\sigma$ for all $L' > L$.*

*Proof.* We denote with $W_{\ell,\cdot i}$ the $i$-th column of $W_\ell$ and with $W_{\ell,i\cdot}$ the $i$-th row of $W_\ell$. We can decompose the trace using the columns $W_{L,\cdot i}$ and rows $W_{L-1,i\cdot}$ in the following way:

$$\mathrm{Tr}\big[G^\top Z_L\big] = \sum_{i=1}^{w_L} \mathrm{Tr}\big[G^\top W_{L,\cdot i}\sigma(W_{L-1,i\cdot} Z_{L-2})\big].$$

The negative contribution is entirely due to the $W_L$ matrix as the application of the activation function yields a non-negative matrix. For this sum there exists some $i^* \in [w_L]$ that maximizes the negative contribution so that for all $i \in [w_L]$:

$$\mathrm{Tr}\big[G^\top \bar{W}_{L,\cdot i^*} \sigma(\bar{W}_{L-1,i^*\cdot} Z_{L-2})\big] \leq \mathrm{Tr}\big[G^\top \bar{W}_{L,\cdot i} \sigma(\bar{W}_{L-1,i\cdot} Z_{L-2})\big]$$

where $\bar{x}$ denotes the normalized vector $\bar{x} = \frac{x}{\|x\|_2}$. We define a new network of depth $L + k$ using the following matrices $\tilde{W}_\ell$:

$$\tilde{W}_{L-1} = \sqrt{\sum_i \|W_{L,\cdot i}\| \|W_{L-1,i\cdot}\|} \begin{pmatrix} \bar{W}_{L-1,i^*\cdot} \\ 0 \\ 0 \\ \vdots \\ 0 \end{pmatrix},$$

$$\tilde{W}_{L+k} = \sqrt{\sum_i \|W_{L,\cdot i}\| \|W_{L-1,i\cdot}\|} \begin{pmatrix} \bar{W}_{L,\cdot i^*} & 0 & 0 & \cdots & 0 \end{pmatrix},$$

$$\tilde{W}_\ell = \begin{pmatrix} 1 & 0 & 0 & \cdots \\ 0 & 0 & 0 & \cdots \\ \vdots & \vdots & \ddots & \vdots \\ 0 & 0 & \cdots & 0 \end{pmatrix}, \quad \ell = L, \ldots, L+k-1,$$

$$\tilde{W}_\ell = W_\ell, \quad \ell = 1, 2, \ldots, L-2$$

We observe that the trace of the new network is lower or equal to the trace of the original network:

$$\begin{aligned}
\mathrm{Tr}\Big[G^\top \tilde{Z}_{L+k}\Big] &= \mathrm{Tr}\Big[G^\top \tilde{W}_{L+k} \sigma(\tilde{W}_{L+k-1,i\cdot} Z_{L+k-2})\Big] \\
&= \mathrm{Tr}\big[G^\top \bar{W}_{L,\cdot i^*} \sigma(\bar{W}_{L-1,i^*\cdot} Z_{L-2})\big] \sum_{i=1}^{w_L} \|W_{L,\cdot i}\| \|W_{L-1,i\cdot}\| \\
&\leq \sum_{i=1}^{w_L} \|W_{L,\cdot i}\| \|W_{L-1,i\cdot}\| \, \mathrm{Tr}\big[G^\top \bar{W}_{L,\cdot i} \sigma(\bar{W}_{L-1,i\cdot} Z_{L-2})\big] \\
&= \sum_{i=1}^{w_L} \mathrm{Tr}\big[G^\top W_{L,\cdot i} \sigma(W_{L-1,i\cdot} Z_{L-2})\big] \\
&= \mathrm{Tr}\big[G^\top Z_L\big].
\end{aligned}$$

The norm of the new network is :

$$\begin{aligned}
\|\tilde{\theta}\|^2 &= \sum_{\ell=1}^{L-2} \|W_\ell\|^2 + 2 \sum_{i=1}^{w_L} \|W_{L,\cdot i}\| \|W_{L-1,i\cdot}\| + k \\
&\leq \sum_{\ell=1}^{L} \|W_\ell\|^2 + k \\
&\leq L + k
\end{aligned}$$

$\square$

### A.5 LOW RANK BIAS

#### A.5.1 WEAK CONTROL

Before presenting our results we will describe a simple lemma that's useful to our proofs.

**Lemma A.4.** *For a depth-L network with $\|\theta^2\| \leq L$ we have that*

$$\prod_{\ell=1}^{L} \|W_\ell\|_F \leq 1$$

*Proof.* This essentially follows from the AM-GM inequality:

$$(\frac{\|\theta\|^2}{L}) = (\frac{1}{L}\sum_\ell \|W_\ell\|_F^2)^{\frac{L}{2}} \geq (\prod_\ell \|W_\ell\|_F^2)^{\frac{1}{2}} = \prod_\ell \|W_\ell\|_F$$

and using the bounded norm assumption

$$1^{\frac{L}{2}} = 1 \geq (\frac{\|\theta\|^2}{L})^{\frac{L}{2}}.$$

$\square$

**Proposition A.5.** *Given that $Tr[G^\top Z_L] \leq -s_0$, for some constant $s_0 > 0$ and $\|\theta\|^2 \leq L$ then for any ratio $p \in (0,1)$ there are at least $(1-p)L$ layers that are approximately rank 1 in the sense that the singular values $s_i$ of $Z_\ell^\sigma$ satisfy*

$$\frac{\sum_{i \geq 2} s_i^2}{\sum_{i \geq 1} s_i^2} \leq 2\frac{\log \|X\|_F + \log \|G\|_F - \log s_0}{pL}. \tag{3}$$

*Proof.* We expand the activations,

$$\frac{\|Z_0^\sigma\|_F^2}{\|Z_L\|_F^2} = \frac{\|Z_{L-1}^\sigma\|_{op}^2}{\|Z_L\|_F^2} \prod_{\ell=1}^{L-1} \frac{\|Z_{\ell-1}^\sigma\|_{op}^2}{\|Z_\ell^\sigma\|_F^2} \prod_{\ell=0}^{L-1} \frac{\|Z_\ell^\sigma\|_F^2}{\|Z_\ell^\sigma\|_{op}^2}.$$

Where the operator norm of a matrix is its largest singular value.

Since $Z_\ell = W_\ell Z_{\ell-1}^\sigma$, we have

$$\|Z_\ell\|_F^2 \leq \|W_\ell\|_F^2 \|Z_{\ell-1}^\sigma\|_{op}^2.$$

So by using lemma A.4,

$$\frac{\|Z_{L-1}^\sigma\|_{op}^2}{\|Z_L\|_F^2} \prod_{\ell=1}^{L-1} \frac{\|Z_{\ell-1}^\sigma\|_{op}^2}{\|Z_\ell^\sigma\|_F^2} \prod_{\ell=0}^{L-1} \frac{\|Z_\ell^\sigma\|_F^2}{\|Z_\ell^\sigma\|_{op}^2} \geq \prod_{\ell=0}^{L-1} \frac{\|Z_\ell^\sigma\|_F^2}{\|Z_\ell^\sigma\|_{op}^2}$$

On the other hand we have that

$$\frac{\|Z_0^\sigma\|_F^2 \|G\|_F^2}{\text{Tr}[G^\top Z_L]^2} \geq \frac{\|Z_0^\sigma\|_F^2}{\|Z_L\|_F^2}$$

since the inner product is always lower than the norm product.

Now by combining the above we get

$$\prod_{\ell=0}^{L-1} \frac{\|Z_\ell^\sigma\|_F^2}{\|Z_\ell^\sigma\|_{op}^2} \leq \frac{\|Z_0^\sigma\|_F^2 \|G\|_F^2}{\text{Tr}[G^\top Z_L]^2}.$$

Taking the log on both sides,

$$\sum_{\ell=1}^{L-1} \log \|Z_\ell^\sigma\|_F^2 - \log \|Z_\ell^\sigma\|_{op}^2 \leq \log \frac{\|Z_0^\sigma\|_F^2 \|G\|_F^2}{\text{Tr}[G^\top Z_L]^2}.$$

By contradiction, we see that for any ratio $p \in (0, 1)$, there can be at most $pL$ layers where

$$\log \|Z_\ell\|_F - \log \|Z_\ell\|_{op} \geq \frac{\log \|X\|_F + \log \|G\|_F - \log s_0}{pL}.$$

That is there is at least $(1 - p)L$ layers where

$$\frac{\sum_{i \geq 2} s_i^2}{\sum_{i \geq 1} s_i^2} = 1 - \frac{\|Z_\ell\|_{op}^2}{\|Z_\ell\|_F^2} \leq 2 \log \|Z_\ell\|_F - 2 \log \|Z_\ell\|_{op} \leq 2 \frac{\log \|X\|_F + \log \|G\|_F - \log s_0}{pL}.$$

$\square$

### A.5.2 STRONG CONTROL ON ALMOST RANK 1 INPUT

The following result shows that if the input of the network is approximately rank-1, here encoded as $uv^T + X$, where $u, v$ are non-negative entry-wise vectors and $X$ is a matrix of small norm $\|X\|_F \leq \epsilon$, then all layers are approximately Rank-1 too at the optimal escape direction.

**Proposition A.6.** *Consider the minimizer $\theta^\star = \arg\min_{\|\theta\|^2 \leq L} \text{Tr}[G^\top Y_\theta(uv^\top + X)]$ where $u, v \in \mathbb{R}^n$, $u, v \geq 0$ entry wise and $\|X\|_F \leq \epsilon$ for some $\epsilon > 0$. Then for all $\ell$ we have:*

$$\frac{\sum_{i \geq 2} s_i^2(W_\ell)}{\sum_{i \geq 1} s_i^2(W_\ell)}, \frac{\sum_{i \geq 2} s_i^2(Z_\ell^\sigma)}{\sum_{i \geq 1} s_i^2(Z_\ell^\sigma)}, \frac{\|Z_\ell^\sigma - Z_\ell\|_F^2}{\|Z_\ell\|_F^2} \leq 8 \frac{\|G\|_F}{s^* - \|G\|_F \epsilon} \epsilon.$$

*Proof.* In the case where the input is only the rank 1 matrix $uv^\top$ we can see that

$$\text{Tr}[G^\top Y_\theta(uv^\top)] = \text{Tr}[v^\top G^\top Y_\theta(u)] = v^\top G^\top Y_\theta(u)$$

and therefore the minimum is achieved when the alignment is maximized:

$$\min_{\|\theta\|^2 \leq L} v^\top G^\top Y_\theta(u) = -\|Gv\| \|u\|.$$

When $\|\theta\|^2 \leq L$ it is true that

$$\|Y_\theta(uv^\top) - Y_\theta(uv^\top + X)\|_F \leq \prod_{\ell=1}^{L} \|W_\ell\|_F \|X\|_F \leq \epsilon$$

as a consequence of the Cauchy-Schwarz inequality.

We can also see that

$$|\text{Tr}[G^\top Y_\theta(uv^\top + X)] - \text{Tr}[G^\top Y_\theta(uv^\top)]| \leq \|G\|_F \epsilon.$$

At the minimum $\theta^\star = \arg\min_{\|\theta\|^2 = L} \text{Tr}[G^\top Y_\theta(uv^\top + X)]$ we observe that

$$\text{Tr}[G^\top Y_{\theta^\star}(uv^\top + X)] \leq \text{Tr}[G^\top Y_{\hat{\theta}}(uv^\top + X)] \leq -\|Gv\| \|u\| + \|G\|_F \epsilon$$

where $\hat{\theta} = \arg\min \text{Tr}\left[G^\top Y_\theta(uv^\top)\right]$.

On the other direction we get

$$\text{Tr}\left[G^\top Y_{\theta^\star}(uv^\top + X))\right] \geq -\text{Tr}\left[G^\top Y_{\theta^\star}(uv^\top)\right] - \|G\|_F \epsilon$$
$$\geq -\|Gv\|\|Y_{\theta^\star}(u)\| - \|G\|_F \epsilon \qquad (4)$$

where we used the Cauchy-Schwarz inequality in the last line.

Combining the two, we get

$$\frac{\|Y_{\theta^\star}(u)\|}{\|u\|} \geq 1 - \frac{2\|G\|_F}{\|Gv\|\|u\|}\epsilon$$

and since $\|\theta\|^2 \leq L$ it is also true that the LHS of the above inequality is upper bounded by 1.

We can also see that

$$\frac{\|Y_\theta^\star(uv^\top + X)\|}{\|uv^\top + X\|} \geq \frac{\|Y_\theta^\star(uv^\top)\| - \epsilon}{\|uv^\top\| + \epsilon} = \frac{\frac{\|Y_\theta^\star(u)\|}{\|u\|} - \frac{\epsilon}{\|u\|\|v\|}}{1 + \frac{\epsilon}{\|u\|\|v\|}}$$

and by using the above inequality we get

$$\frac{\|Y_\theta^\star(uv^\top + X)\|}{\|uv^\top + X\|} \geq 1 - \frac{\frac{2\|G\|_F}{\|Gv\|\|u\|} - 2\frac{1}{\|u\|\|v\|}}{1 + \frac{\epsilon}{\|u\|\|v\|}}\epsilon \geq 1 - 2\left(\frac{\|G\|_F}{\|Gv\|\|u\|} + \frac{1}{\|u\|\|v\|}\right)\epsilon.$$

Now we can expand the activations,

$$\frac{\|Y_\theta^\star(uv^\top + X)\|}{\|uv^\top + X\|} = \prod_{\ell=1}^{L} \frac{\|Z_\ell\|_F}{\|Z_{\ell-1}^\sigma\|_F} \frac{\|Z_{\ell-1}^\sigma\|_F}{\|Z_{\ell-1}\|_F}$$

and using the fact that $\prod_\ell \|W_\ell\|_F \leq 1$

$$\prod_{\ell=1}^{L} \frac{\|W_\ell Z_{\ell-1}\|_F}{\|W_\ell\|_F \|Z_{\ell-1}^\sigma\|_F} \frac{\|Z_{\ell-1}^\sigma\|_F}{\|Z_{\ell-1}\|_F} \geq \prod_{\ell=1}^{L} \frac{\|W_\ell Z_{\ell-1}\|_F}{\|Z_{\ell-1}^\sigma\|_F} \frac{\|Z_{\ell-1}^\sigma\|_F}{\|Z_{\ell-1}\|_F}. \qquad (5)$$

We split the norm of the activations using the inequalities

$$\frac{\|W_\ell Z_{\ell-1}^\sigma\|_F}{\|W_\ell\|_F \|Z_{\ell-1}^\sigma\|_F} \leq \frac{\|W_\ell\|_{op}\|Z_{\ell-1}^\sigma\|_F}{\|W_\ell\|_F \|Z_{\ell-1}^\sigma\|_F} = \frac{\|W_\ell\|_{op}}{\|W_\ell\|_F}$$

and

$$\frac{\|W_\ell Z_{\ell-1}^\sigma\|_F}{\|W_\ell\|_F \|Z_{\ell-1}^\sigma\|_F} \leq \frac{\|W_\ell\|_F \|Z_{\ell-1}^\sigma\|_{op}}{\|W_\ell\|_F \|Z_{\ell-1}^\sigma\|_F} = \frac{\|Z_{\ell-1}^\sigma\|_{op}}{\|Z_{\ell-1}^\sigma\|_F}$$

so we get

$$\prod_{\ell=1}^{L} \min\left\{\frac{\|W_\ell\|_{op}}{\|W_\ell\|_F}, \frac{\|Z_{\ell-1}^\sigma\|_{op}}{\|Z_{\ell-1}^\sigma\|_F}\right\} \frac{\|Z_{\ell-1}^\sigma\|_F}{\|Z_{\ell-1}\|_F} \geq 1 - 2\left(\frac{\|G\|_F}{\|Gv\|\|u\|} + \frac{1}{\|u\|\|v\|}\right)\epsilon.$$

By squaring and rearranging the terms we get for the first ratio:

$$\frac{\sum_{i\geq 2} s_i^2(W_\ell)}{\sum_{i\geq 1} s_i^2(W_\ell)} \leq 4\left(\frac{\|G\|_F}{\|Gv\|\|u\|} + \frac{1}{\|u\|\|v\|}\right)\epsilon$$

which further simplifies to

$$\frac{\sum_{i\geq 2} s_i^2(W_\ell)}{\sum_{i\geq 1} s_i^2(W_\ell)} \leq 8\frac{\|G\|_F}{\|Gv\|\|u\|}\epsilon.$$

Using inequality 4 we observe that:

$$\|Gv\|\|u\| \geq s^\star - \|G\|_F\epsilon$$

and hence

$$\frac{\sum_{i\geq 2} s_i^2(W_\ell)}{\sum_{i\geq 1} s_i^2(W_\ell)} \leq 8\frac{\|G\|_F}{s^\star - \|G\|_F\epsilon}\epsilon.$$

We proceed similarly for the singular values of $Z_\ell$.

For the matrices $Z_\ell$ and $Z_\ell^\sigma$ we note that their Frobenius inner product is zero, so

$$\|Z_\ell\|_F^2 = \|Z_\ell^\sigma\|_F^2 + \|Z_\ell - Z_\ell^\sigma\|_F^2$$

Re-arranging gives the inequality

$$\frac{\|Z_\ell^\sigma - Z_\ell\|_F^2}{\|Z_\ell\|_F^2} \leq 8\frac{\|G\|_F}{s^\star - \|G\|_F\epsilon}\epsilon.$$

$\square$

### A.5.3 STRONG CONTROL

We can combine the above two statements to show that at the maximum escape speed, the final layers will be almost rank-1. To prove that we first apply proposition A.5 to find a layer $\ell_0$ that is almost rank-1. We need an additional lemma that ensures that we can select non-negative singular vectors for the largest singular value of the activation $Z_{\ell_1}$. Then we can apply proposition A.6 to conclude that all layers $\ell \geq \ell_0$ will be approximately rank-1.

**Lemma A.7.** *For $A \in R^{m\times n}$ with non-negative entries and $s_1$ its largest singular value we can find right and left singular values $u_1, v_1$ for $s_1$ which are non-negative entry-wise.*

*Proof.* The right singular vector for $s_1$ satisfies

$$A^\top A u_1 = s_1 u_1$$

and since $A$ is non-negative $A^\top A$ is also non-negative. We can now apply an extended version of the Perron-Frobenius theorem for non-negative matrices to select the eigenvector $u_1 \geq 0$ entry-wise. Now we select

$$v_1 = \frac{Au_1}{s_1}$$

which is a left singular vector as it satisfies $Av_1 = s_1 u_1$ and since $A, u$ are non-negative, $v$ is also non-negative. $\square$

**Theorem A.8** (Theorem 3.1 in the main). *Consider an optimal escape direction*

$$\theta^\star = \underset{\|\theta\|^2 = L}{\arg\min} \operatorname{Tr}\big[G^\top Y_\theta\big]$$

*with optimal speed* $s^* = min_{\|\theta\|^2 = L} \operatorname{Tr}\big[G^\top Y_\theta\big]$, *then for all layers* $\ell$ *we have:*

$$\frac{\sum_{i\geq 2} s_i^2(W_\ell)}{\sum_{i\geq 1} s_i^2(W_\ell)}, \frac{\sum_{i\geq 2} s_i^2(Z_\ell^\sigma)}{\sum_{i\geq 1} s_i^2(Z_\ell^\sigma)}, \frac{\|Z_\ell^\sigma - Z_\ell\|_F^2}{\|Z_\ell\|_F^2} \leq 8 \frac{c}{s^* - c\ell^{-\frac{1}{2}}} \ell^{-\frac{1}{2}}$$

*where* $c = \sqrt{2}\|X\|_F\|G\|_F \sqrt{\log \|X\|_F + \log \|G\|_F - \log s^*}$.

*Proof.* We denote $Y_{\ell_2:\ell_1}(X) = W_{\ell_2}\sigma(W_{\ell_2-1}...\sigma(W_{\ell_1}X)...)$ the network when only the layers from $\ell_1$ to $\ell_2$ are applied.

Using proposition A.5 we can find layers $\ell_0 < \ell_1 < ... < \ell_n \leq L$ that satisfy 3 and are approximately rank-1. We can select the minimum of those, $\ell_0$.

Because the argument is valid for at least $(1-p)L$ of the $L$ total layers, the earliest layer $\ell_0$ must occur on or before the $pL$-th layer.

It is true that $Z_{\ell_0}^\sigma$ is non-negative entry-wise and so we can apply lemma A.7 to find non-negative singular vectors $u_1, v_1$ that additionally satisfy

$$\|Z_{\ell_0}^\sigma - s_1(Z_{\ell_0}^\sigma)u_1 v_1^\top\|_F^2 = \sum_{i=2}^r s_i^2 \leq 2\|Z_{\ell_0}^\sigma\|_F^2 \frac{\log \|X\|_F + \log \|G\|_F - \log s^*}{pL}$$

$$\leq 2\|X\|_F^2 \frac{\log \|X\|_F + \log \|G\|_F - \log s^*}{pL}.$$

We now use the fact that for the layers $\ell \geq \ell_0$ we are at the optimal escape direction $\theta^*$.

$$\theta^* = \underset{\|\theta_{L:\ell_0+1}\|^2 = L - \ell_0}{\arg\min} \operatorname{Tr}\big[G^\top Y_{L:\ell_0+1}(Z_{\ell_0}^\sigma)\big].$$

We can do that since:

$$\min_{\|\theta\|^2 = L} \operatorname{Tr}\big[G^\top Y_\theta\big] \leq \min_{\|\theta_{L:\ell_0+1}\|^2 = L - \ell_0} \operatorname{Tr}\big[G^\top Y_{L:\ell_0+1}(Z_{\ell_0}^\sigma)\big]$$

We can now apply proposition A.6 on the sub-network $Y_{L:\ell_0+1}$. For all layers $\ell \geq \ell_0$ we have that:

$$\frac{\sum_{i\geq 2} s_i^2(W_\ell)}{\sum_{i\geq 1} s_i^2(W_\ell)}, \frac{\sum_{i\geq 2} s_i^2(Z_\ell^\sigma)}{\sum_{i\geq 1} s_i^2(Z_\ell^\sigma)}, \frac{\|Z_\ell^\sigma - Z_\ell\|_F^2}{\|Z_\ell\|_F^2} \leq 8 \frac{c}{1 - \frac{c}{\sqrt{pL}}} \frac{1}{\sqrt{pL}}$$

where $c = \frac{\|X\|_F\|G\|_F}{s^*} \sqrt{2 \log \frac{\|X\|_F\|G\|_F}{s^*}}$.

We see that, since $p \in (0,1)$ was chosen arbitrarily, we can choose $p = \frac{\ell}{L}$ for any $\ell$. Then since $\ell_0 \leq pL = \ell$ the above inequality will hold for any $\ell$-th layer with $\ell > c^2$.

$\square$

# B  MNIST TRAINING DETAILS

We train a 6-layer fully connected neural network (multilayer perceptron, MLP) without biases on the MNIST dataset, using the cross-entropy loss. The network comprises one input layer, four hidden layers, and one output layer. Each hidden layer contains 1000 neurons. The weight matrices have the following dimensions:

- **Input layer:** $W_1 \in \mathbb{R}^{784 \times 1000}$
- **Hidden layers:** $W_i \in \mathbb{R}^{1000 \times 1000}$ for $i = 2, 3, 4, 5$
- **Output layer:** $W_6 \in \mathbb{R}^{1000 \times 10}$

The weights are initialized from a normal distribution with mean 0 and standard deviation $1/1000$.

We train the model for 1000 epochs using a batch size of 32. The learning rate at each step is adjusted dynamically according to:

$$\mathrm{lr}(t) = \frac{10}{\|\theta(t)\|^4}$$

where

$$\|\theta(t)\|^2 = \sum_{i=1}^{6} \|W_i(t)\|_F^2$$

and $\|\cdot\|_F$ denotes the Frobenius norm.

Each MNIST image $x$ is normalized using the dataset-wide mean $\mu$ and standard deviation $\sigma$ of the pixel values:

$$x \mapsto \frac{x/255 - \mu}{\sigma}$$

This standardization ensures that the input distribution has approximately zero mean and unit variance, which helps stabilize training.

A more complete picture of how the singular values of the weight matrices evolve during training is presented in 3.

We repeated the same experiment with depth-4 fully connected network and we report our findings in figure 4.

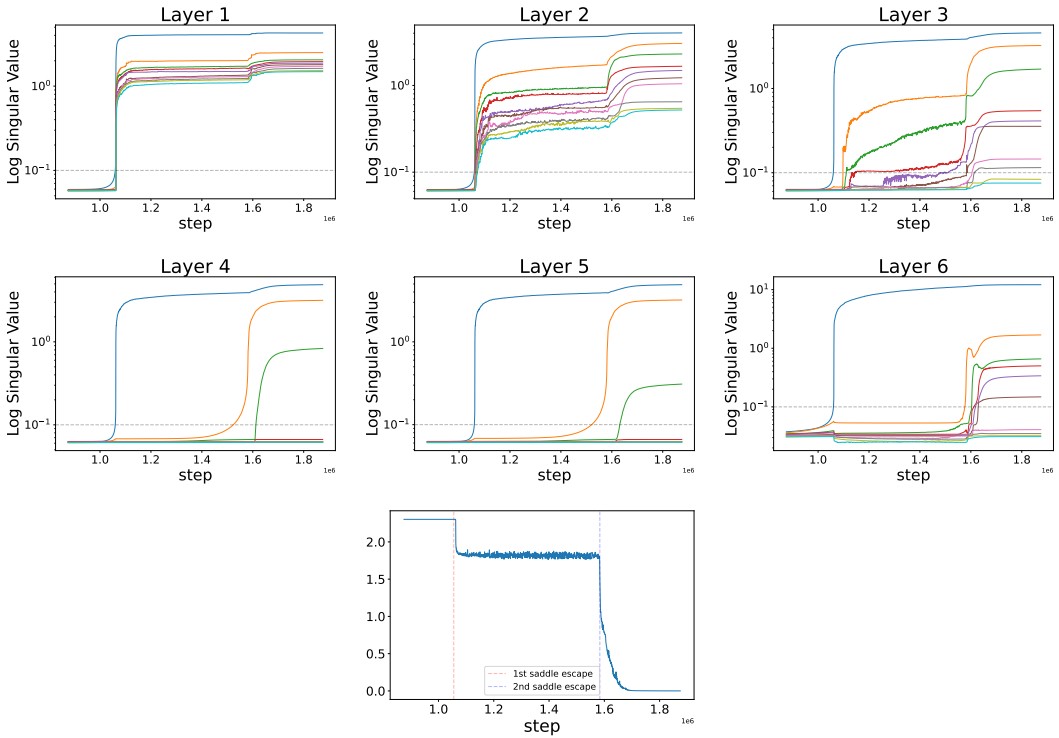

Figure 3: **Deeper layers show a stronger bias toward low-rank structure than earlier layers on MNIST. Top two rows:** Top 10 singular values of the weight matrices for layers 1–6 including input and output layer over training time. **Bottom:** Training loss trajectory on MNIST.

## C   CIFAR-10 TRAINING DETAILS

We train a 6-layer fully connected neural network without biases on the CIFAR-10 dataset, using the cross-entropy loss. The network comprises one input layer, four hidden layers, and one output layer. Each hidden layer contains 1000 neurons. The weight matrices have the following dimensions:

- **Input layer:** $W_1 \in \mathbb{R}^{1024 \times 1000}$
- **Hidden layers:** $W_i \in \mathbb{R}^{1000 \times 1000}$ for $i = 2, 3, 4, 5$
- **Output layer:** $W_6 \in \mathbb{R}^{1000 \times 10}$

The weights are initialized from a normal distribution with mean 0 and standard deviation $1/1000$.

We train the model for 5000 epochs using a batch size of 32. The learning rate at each step is adjusted dynamically according to:

$$\text{lr}(t) = \frac{40}{\|\theta(t)\|^4}$$

where

$$\|\theta(t)\|^2 = \sum_{i=1}^{6} \|W_i(t)\|_F^2$$

and $\|\cdot\|_F$ denotes the Frobenius norm.

Each CIFAR-10 image $x$ is normalized using the dataset-wide mean $\mu$ and standard deviation $\sigma$ of the pixel values:

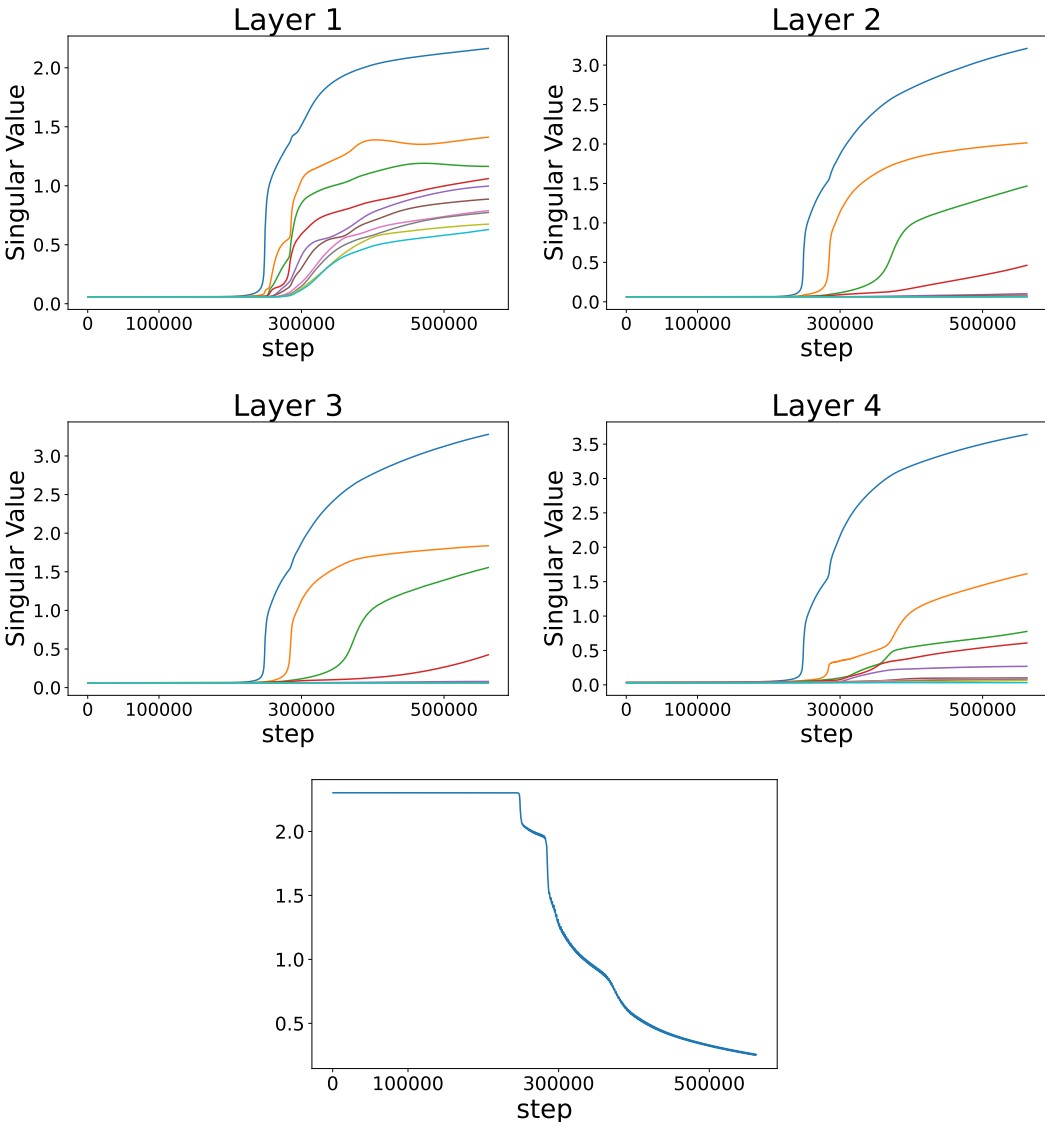

Figure 4: **Depth-4 MLP with small initialization on MNIST. Top two rows:** Top 10 singular values of the weight matrices for layers 1–4 including input and output layer over training time. **Bottom:** Training loss trajectory on MNIST.

$$x \mapsto \frac{x/255 - \mu}{\sigma}$$

A more complete picture of how the singular values of the weight matrices evolve during training is presented in 5.

We repeated the same experiment with depth-4 fully connected network and we report our findings in figure 6. For the depth-4 network we used the learning rate $\mathrm{lr}(t) = \frac{0.01}{\|\theta(t)\|^4}$ and 10000 epochs to ensure convergence.

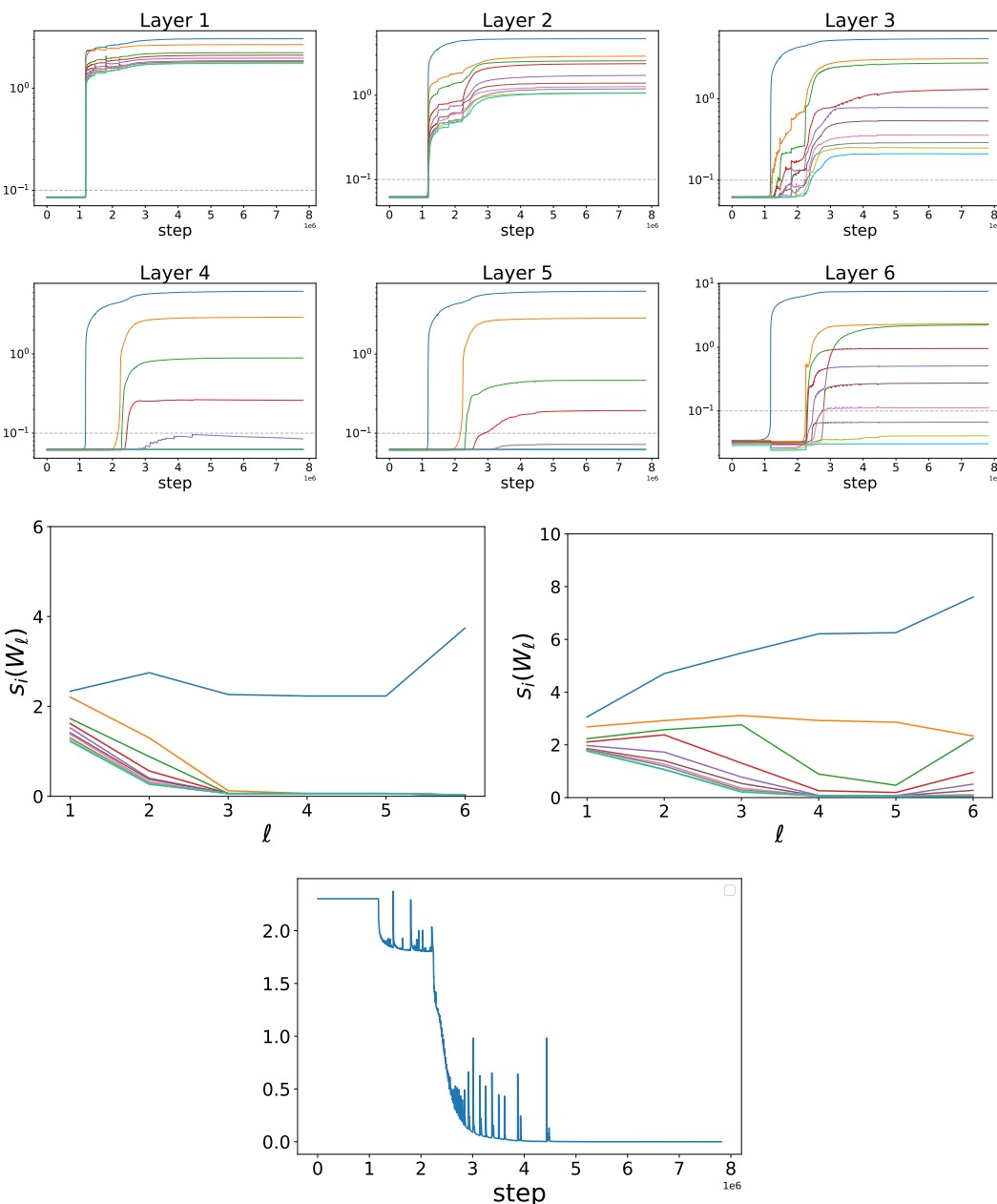

Figure 5: **Deeper layers show a stronger bias toward low-rank structure than earlier layers on CIFAR-10. Top two rows:** Top 10 singular values of the weight matrices for layers 1–6 including input and output layer over training time in logarithmic scale. **Third Row**: Top 10 singular values of the weight matrices $W_\ell$ across layers 1–6 (including input/output). **Left**: After the first saddle point escape. **Right**: At the end of training. **Bottom:** Training loss trajectory on CIFAR-10.

# D    SUPPORTING MATERIAL FOR SECTION 4

## D.1    FINDING THE MAXIMAL RANK-ONE ESCAPE SPEED

Picking up the argument from the proof sketch of Example 1, we have a network function equal to $f(X) = \pm\sigma(W_1 X)$, where $W_1 = [\cos(\phi), \sin(\phi)]$ and the sign is chosen to give a positive escape speed. Applied to the dataset of Example 1 and noting that at most four points will have nonzero

function value at a given time, one finds an escape speed is equal to

$$s = \left| \cos\left(\xi + \frac{\pi}{4}\right) - \cos(\xi) + \cos\left(\xi - \frac{\pi}{4}\right) - \cos\left(\xi - \frac{\pi}{2}\right) \right|, \tag{6}$$

where $\xi = \phi \mod(\frac{\pi}{4})$. See Figure 7 for a depiction of this periodic function. Its maximal value of $s = \sqrt{2} - 1$ falls at multiples of $\frac{\pi}{4}$.

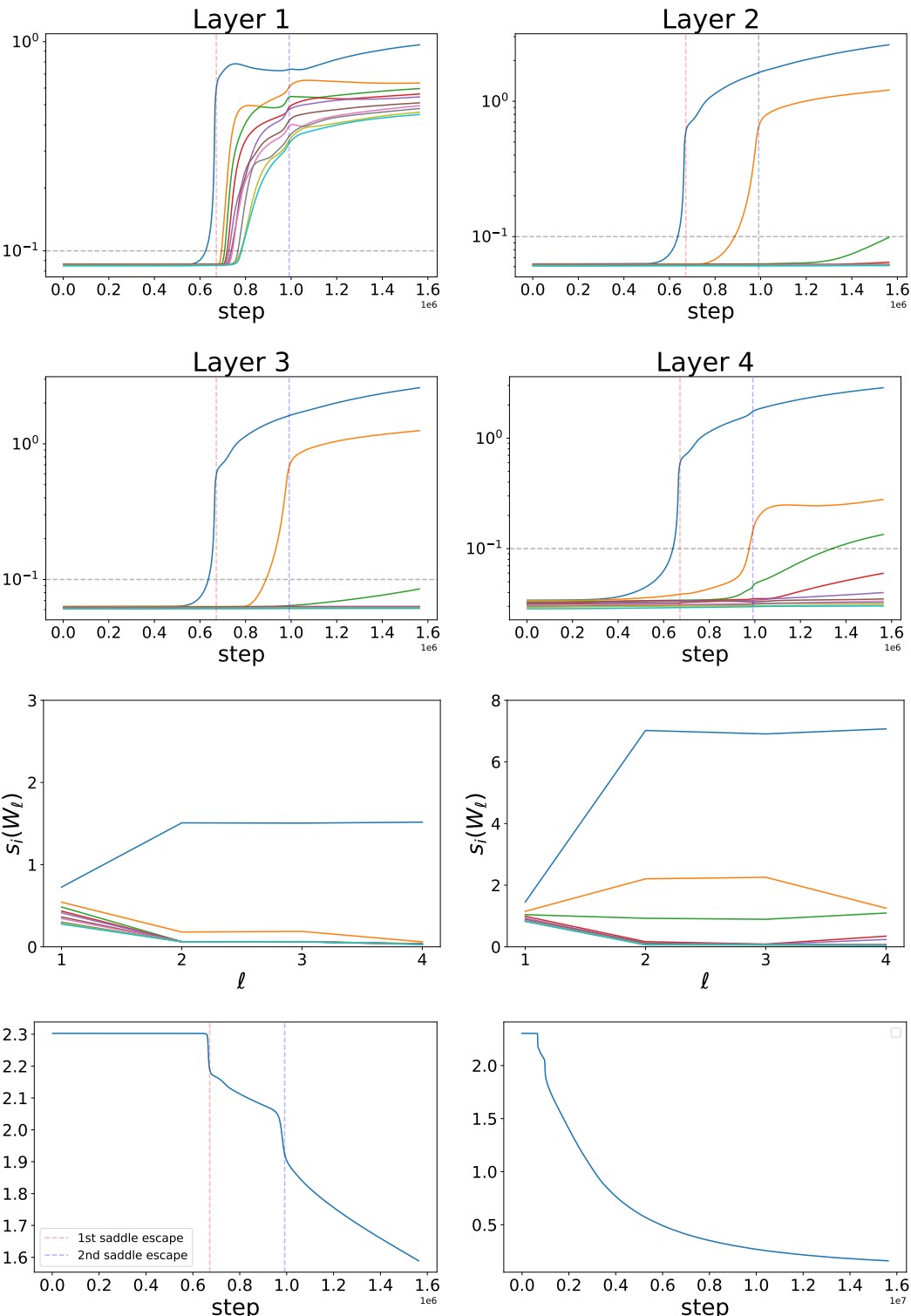

Figure 6: **Depth-4 MLP with small initialization on CIFAR-10. Top two rows:** Top 10 singular values of the weight matrices for layers 1–4 including input and output layer over training time in logarithmic scale during the initial 1000 epochs. **Third Row**: Top 10 singular values of the weight matrices per layer $\ell$ for layers 1–4 including input and output layer after the first saddle escape and at the end of training. **Bottom:** Training loss curves on CIFAR-10. **Left:** The initial 1000 epochs. **Right:** The full training run of 10000 epochs.

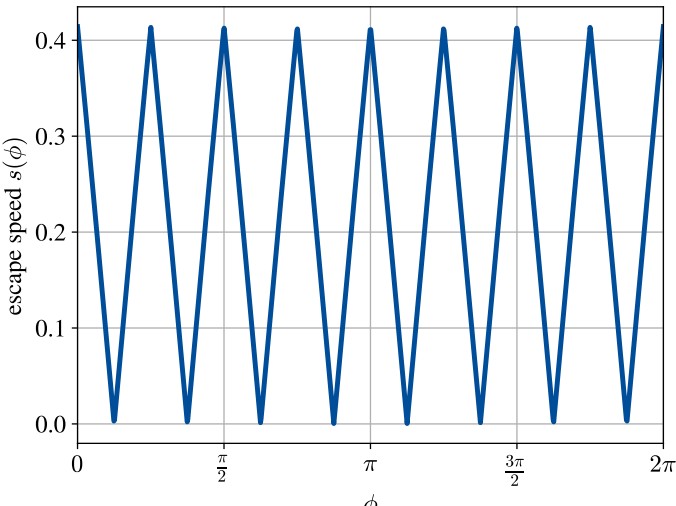

Figure 7: Visualization of Equation 6.

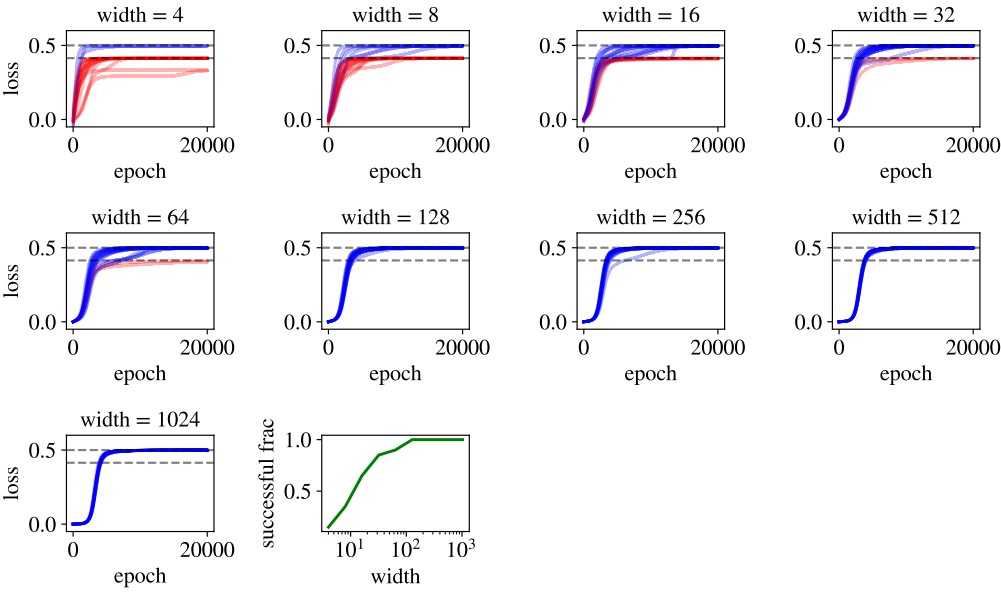

Figure 8: Visualization of all training runs of projected gradient descent on Example 1. This plot shows all training runs in the experiment of Figure 2.

