# OpenReview forum: "Saddle-To-Saddle Dynamics in Deep ReLU Networks: Low-Rank Bias in the First Saddle Escape"
_ICLR.cc/2026/Conference — ICLR 2026 Poster_

### Official Review · Reviewer_Y4Ze · 2025-10-30

**Soundness:** 3
**Presentation:** 3
**Contribution:** 3
**Rating:** 6
**Confidence:** 3

**Summary:**

The paper provides an analytical study of the low-rank bias observed in the first escape direction in multi-layer ReLU networks. In fact, when the network is initialized with small weights, it needs to escape from the saddle point present at the origin, and the direction of this escape was previously shown to be biased towards low ranks. The paper formally proves that the escape direction is biased to have approximately rank 1 in deeper layers. A simple neural network trained on MNIST validates the theory and an analytical counterexample is provided to confirm that the rank of the optimal escape direction can be higher than one and thus an approximate notion of rank one is necessary.

**Strengths:**

- The background literature in Section 1 is very well presented. The reader gets a clear picture of this line of research, how previous works contributed to it, and how this paper fits into the narrative.
- The main results (Theorem 3.1. and Proposition 3.2.) are novel and provide an interesting theoretical advancement in the context of understanding saddle-to-saddle dynamics.

**Weaknesses:**

Although both theoretical and experimental examples were given to corroborate the claim, the latter are very limited in scope, with only one experiment performed on one simple architecture and dataset. I recognize that this is mainly a theoretical work, but a more robust numerical evaluation would have been welcome to validate the theory.

The results mainly concern the specific case of standard ReLU MLPs with no regularization (otherwise the loss is not homogeneous) and trained with gradient flow. These are standard assumptions in the literature but their strict nature limits the scope of the results.

I believe that the authors could improve the readability of the paper for a reader that is not an expert in this line of research:
- The approximation around the origin of Eq. (1) should be derived in the text, in the appendix, or a reference where such derivation is made should be referenced.
- The same holds for the homogeneity of the loss at L173. A good reference plus an intuition on where it comes from would be enough I think.
- The derivation of the equation at L180-181 (which should be numbered) in the appendix should be referenced in the main text. This should hold for all other derivations in the appendix.

### Minor weaknesses
- Please provide full forms of all abbreviations before using them, e.g. DNN (L025), NTK (L029), CNN (L076)

**Questions:**

- At L171 it is claimed that "since the ReLU is not differentiable, neither is the loss at the origin, so that we cannot use the traditional strategy of approximating L0 by a polynomial.". How is the approximation of Eq. (1) justified in light of this fact?
- How much do the authors think the result extend (if they do) to the case of regularized losses or other activation functions?

---

> ### Author Response · Authors · 2025-11-20
>
> Thank you for the thoughtful review. Regarding the weaknesses you mention:
> - More experiments: We will add more experiments in the appendix, e.g. CIFAR (we mighht not be able to do it by the deadline for the rebuttals, but definitely for the final version).
> - Note that the main results (that the optimal escape directions are low rank) is independent of GF/GD. We only use GF in Section 2 to motivate why optimal escape directions are important. This would be easy to extend to GD, but because our goal is to give an intuition of the dynamics for the reader we preferred keep the simplest setting that illustrates this, especially if this added complexity does not add any particularly new insights. Regarding regularization, do you mean L2-regularization? Actually, a bias towards low Bottleneck rank was first proven with L2 regularization (Jacot 2023a, 2023b), the goal of this paper is to show that this bias also arises without L2 regularization.
>
> We agree with the points you make about clarity, we have added a derivation of equation 1 in the appendix with references. We will also introduce the abbreviations.
>
> Regarding your questions:
> 1. The outputs of the network are not differentiable w.r.t the weights, but we are taking the Taylor approximation of the loss w.r.t the outputs of the network which is differentiable, e.g. the MSE loss $C(Y)=\frac{1}{N}\|Y-Y^*\|^2_F$ which is quadratic polynomial.
> 2. As mentioned earlier, this paper extends previous works of Bottleneck rank bias with L2 regularized networks to the unregularized case, so we expect similar structures to emerge. Note that with L2 regularization, the saddle at the origin becomes a local minima, so one cannot take an infinitely small initialization anymore. This results could be relatively easily extended to any homogeneous activation function (which essentially leaky ReLUs), or possibly activations with other degrees of homogeneity (such as the ReLU squared), but it would be difficult to extend it to non-homogeneous nonlinearities.

---

### Official Review · Reviewer_Bqc1 · 2025-10-30

**Soundness:** 3
**Presentation:** 2
**Contribution:** 3
**Rating:** 6
**Confidence:** 3

**Summary:**

This paper gives a description of the saddle at the origin in deep ReLU networks, and the possible escape directions that GD could take as it escapes this first saddle. It were shown that the optimal escape directions feature a low-rank bias that gets stronger in deeper layers, and the optimal escape speed is non-decreasing in depth. A simple dataset is presented whose optimal escape direction is rank two for the first weight matrix. Finally, a few conjectures/hypotheses describing the complete Saddle-to-Saddle dynamics are discussed.

**Strengths:**

1. The theoretical analysis seems sound, although I did not check the details of proof.
2. Both optimal escape direction and speed are analysed.
3. Figure 1 intuitively illustrates the saddle escape behavior and the low-rank bias feature on the MNIST dataset.

**Weaknesses:**

1. In experiments, the low-rank bias feature is only demonstrated on the MNIST dataset. Results on more datasets are preferred.
2. The motivation is not clear enough. Why should we study the saddle-to-saddle regime for very small initializations? In practice, the weights are initialized with, e.g. Kaiming initialization, which may not fall into the very small initialization regime.
3. What is the practical implications of low-rank bias feature for network training? Does it mean pathological loss landscape? How to use it to escape the saddle more effectively and efficiently?
4. The complete Saddle-to-Saddle dynamics are only given as hypotheses without further proofs or empirical evidences. The main technical contribution is focused on the first saddle.

**Questions:**

see Weaknesses.

---

> ### Author Response · Authors · 2025-11-20
>
> Thank you for your review and interesting questions:
> 1. We will do some experiments on CIFAR and add them to the appendix, we expect very similar results. We might not have time to finish these experiments before the deadline for updating the paper though.
> 2. There are multiple reasons to study the Saddle-to-Saddle regime:
>     - First, the initialization tradeoff can be summarized as: large initialization speeds up training (see exponential convergence in the NTK regime) but at the cost of less sparsity/implicit bias. In practice, one therefore wants to take an init. that is not too small, not too large. The advantage is that theory is not affected by the slow down, so we can try to understand what the smallest dynamics (with the most sparsity) looks like.
>     - The impact of saddles, and how they lead to rank sparsity can still be felt for pretty large initialization (in linear networks as soon as the initialization is small enough to leave the NTK regime).
>     - The transition between the regime unaffected by saddles (NTK regime) and the saddle-to-saddle regime can shift depending on the task. In linear networks, if the task is naturally low rank, the saddle-to-saddle regime starts for larger initializations (Tu et al, 2024).
>     - The recent trend is to use initializations such as Mu-P which are smaller than Kaiming.
> 3. We think it is a bit too early to give some concrete practical advice, but this suggests some interesting directions. First it illustrates the tradeoff of saddles: they slow-down training but conversely also induce some useful implicit sparsity bias that could help generalization. This suggests one should choose an initializations scale that is just small enough to start to see plateaus emerging, but not too small. An interesting future direction is to search for training methods that allow for faster escape of saddles while keeping their implicit bias, thus sidestepping this tradeoff. The rescaling of the learning rate we use in Section 2.1 already helps with that, but it only works at the saddle at the origin, not at the other saddles. It is possible that Adam is good because it has a similar effect at all saddles.
> 4. Yes, we are still a long way from proving the full saddle-to-saddle dynamics in deep ReLU networks, but this paper makes one of the first significant steps towards this goal.

---

### Official Review · Reviewer_bYb4 · 2025-11-01

**Soundness:** 3
**Presentation:** 4
**Contribution:** 3
**Rating:** 8
**Confidence:** 3

**Summary:**

Training of neural networks can happen in two main regimes: lazy and rich, the latter further split into an active and a mean-field regime.
This paper studies the early training dynamics of deep ReLU networks initialized with small weights, focusing on the active training regime, specifically the saddle-to-saddle phase where gradient flow (GF) escapes from the origin saddle in parameter space.
After introducing a *localized loss* approximating the true loss at small scale, norm and spherical dynamics are separated with the latter evolving faster and identifying escape directions, which are minima of the loss constrained to a sphere and therefore points where the norm dynamics can take the lead.
The theoretical results of the paper revolve around conditions imposed on the parameter at escape directions, in particular a low rank bias (seen through singular values of weights and activations matrices) and the collapse of inner activation layer to identity layer, with both these facts amplified with depth of the layer.

A validation numerical experiments is provided and a counter-example is given to disprove a conjecture that the escape direction is always rank one and, finally, the authors propose conjectures about saddle-to-saddle dynamics.

**Strengths:**

The submission appears technically sound, with a mathematically precise yet elegant formalism. The main claims are well supported by both theoretical proofs in the appendix and numerical validations (though not extensive).
The paper is clearly written and well structured, conveying a complete and coherent message.
While the authors do not provide code, the experimental setup is described in sufficient detaisl in Appendix B to allow reproduction, except for the unspecified method used to compute the top 10 singular values, which I think should be precised.
Overall, I have a positive impression of this work and consider it to be of high quality.

The analysis of where the rank-one layer are located, along with the proposal of a half-bottleneck structure, is particularly insightful as it helps reveal (during training) where the network’s interesting computations occur and therefore help with interpretability.

With the insights provided by the paper, there are clear ways forward, specifically, extending the analysis beyond the first saddle escape and exploring how learning might understood iteratively across subsequent saddles.
These potential research paths are further reinforced by the conjectures presented in the discussion section.

**Weaknesses:**

The paper presents a descriptive rather than prescriptive analysis and combined with the theoretical treatment, it limits the immediate practical consequences but the approach is appropriate given the paper’s objectives.
Also, the experimentation are limited in scope and the setup on MNIST seems a bit specific (see the related question below).
I think the paper could benefit from a stronger connection to practical implications, even long term, to help engage a casual reader: although the introduction situates the work well within related literature, the paper overall offers few explicit takeaways or long-term motivations for practitioners. (see last question)

Limitations are:
- The setup does not include biases
- The main result applies for optimal escape directions, however it is not guaranteed that the found escape direction will be optimal under gradient flow (though with the nuance that another result, proposition 3.4 which applies more broadly)
- The formalism is continuous while in practice discrete gradient descent is used
- Emphasis on deep networks:
    - the condition $l>c^2$ is proven to hold on all but finitely many initial layers; which is not a clear practical information
    - the bound is in $l^{\frac{1}{4}}$ which is not so strong (ratio $\approx 3$ for depth $l=100$), which is nuanced by experiments where the effect is clearly visible

## Minor and additional feedback
- line 453: "is greedily searched by first searching among BN-rank one functions" consider rephrasing to avoid repetition e.g. "is greedily search, first among BN-rank one functions"
- line 96: the word "step" is a bit ambiguous, something like "Dynamics with multiple saddle-to-saddle sequences" could be clearer.
- line 109: consider changing "activation" to "preactivation"
- notation overlap in sections 2.1 and 2.2: $s$ denotes both the reparamtrized time and the escape speed
- line 203 & 239: in my understanding this dynamics holds because $\|\theta(t)\|$ evolves more slowly than $\overline\theta(t)$ as it can be seen in equations lines 190. If this is correct, I would recommend precising it explicitly somewhere.
- theorem 3.1 is named A.8 is appendix
- line 323: appendix 3 but appendices are labelled with letters
- line 427: probably you meant "escape speed" instead of "escape direction"

**Questions:**

I have a few questions:
1. **On the parameter norm $r$ of line 228**: can $r$ be estimated to know the scale at which the approximation of GF by the localized loss ceases to hold ?
1. **On the MNIST experiment**: $1000$ hidden neurons are used in each of the 6 layers. In my experience,this level of expressivity exceeds what is needed to fit MNIST with an MLP.
Is there a particular reason for choosing this large width?
More generally, the training setup seems carefully tuned: many training steps ($1000$ epochs of relatively small batches), learning rate scheduling, input standardization.
How necessary are these design choices ?
Could you comment on the sensitivity of the results to the training details ? (Except the initialization scale since it is the focus of the paper)
1. **On figure 3**: what do you observe if you plot $\frac{s_i}{s_1}$ i.e. the ratio of each singular values by the highest one ?
This ratio plot might make it easier to visualize the dominance of the top singular value discussed in the paper.
1. **On conjecture 4**: the paper seems to be a step toward a full characterization of incremental learning in ReLU networks beyond the first saddle.
In your view, what are the most promising long-term practices this line of work could lead to, e.g. improved interpretability via hierarchical learning, overfitting diagnostics, redundancy-based architecture search, or new training strategies that separately address the spherical and explosion phases (as in Kunin et al., 2025)?

---

> ### Author Response · Authors · 2025-11-20
>
> Thank you for your detailed review and the many interesting questions.
>
> About descriptive vs prescriptive: A possible prescription would be that the initialization should be small enough to lead to a low rank bias but large enough to not get stuck at the saddle for too long. Our analysis can capture the second aspect, but the first aspect needs to be understood through the lens of generalization, which remains very hard to quantify. We thus feel like it is a bit too early to be giving concrete advice.
>
> It miight also be possible to completely sidestep this tradeoff by designing better training methods, such as an adaptive learning rate, which should help escape the first saddle. The rescaled learning rate we propose only works for the first saddle, but this could be solved with another scaling in a follow up work.
>
> Regarding the limitations you mention:
> - Adding bias makes the network non-homogeneous, which makes the analysis 'uglier', but it should have essentially no impact (for example Proposition 3.4 could be adapted to handle bias, and it would imply that the bias are zero at most layers). The proof techniques are directly inspired from the Bottleneck rank which works with bias.
>
> - We agree that our biggest assumption is that GF finds the optimal escape direction, we see two possible ways forward: either prove that wide enough networks always find this optimal escape direction (but this might require a very large width in the worst case), or show that the half-bottleneck structure extends to locally optimal escape directions. Both of these would agree with what we observe empirically.
>
> - Our main result is really the half-bottleneck structure at the optimal escape direction, which is not a dynamical result, we only do a simple GF analysis in section 2 to motivate the analysis of these optimal escape directions, but a similar picture should emerge for GD. There is extensive literature on convergence of GD to so-called KKT points (which are essentially the same thing as escape directions) which applies almost directly to our setting, we will add some references.
>
> - The large depth is indeed crucial to our analysis, but it does seem that our analysis is not quite tight in practice (or only tight in the worst case), as we observe a half-bottleneck structure for much smaller depth than what our theory predicts. Also this half-bottleneck structure is often much sharper in the sense that most layers are exactly low rank, not approximately so. But it is possible that one would need assumptions on the data to prove faster rate (if our bound is tight in the worst case).
>
> We agree that our analysis is more qualitative than quantitative (i.e. all but finitely many layers). Though note that we do have a formula for the constant $c$. But because we believe that the rates could potentially still be improved, we did not focus too much on the constants.
>
> Thanks a lot for poiting to these typos and errors, we will make corrections.
>
> Regarding your questions:
> - This is a good point, we have changed the definition of the escape time to be the time where the norm of the outputs is $r$ (instead of the parameter norm), one can then simply take $r=0.1$ for example to get a concrete approximation of the escape time.
> - This a very good question, sadly we do not quite have the data to answer it completely. Since the experiments were there to illustrate the theoretical results, we have optimized for the 'cleanest' curves and separation of the plateaus, and for this reason we have used very large widths and very small initializations. Training becomes very unstable at such small initializations and so adaptive learning rates become almost necessary. To be honest, it does appear that it is 'harder' to reach the saddle-to-saddle regime in ReLU networks than linear networks, and that it is more sensitive to hyper-parameters, but we cannot really give a definite answer yet.
> - That's an interesting idea, but this would probably make things more confusing, because the second and later singular value would first go down before going up, which could give the wrong impression (all singular values are generally increasing, the first singular value just increases faster). We have simply switched to a log scale on the y-axis, which helps see what is happening before the first escape.
> - These are all very interesting prospects indeed. At a higher level, our goal is first to identify the type of sparsity that implicitly emerges in deep ReLU networks and how it can be affected by the architecture (for example we expect this bottleneck rank bias to vanish with residual connections). It also remains open whether it is necessary to get stuck at a saddle to reap its sparsiity benefits? The potential practical impacts will probably depend a lot on the answer to these high level questions.

---

> > ### Comment · Reviewer_bYb4 · 2025-11-25
> >
> > Thank you for your honest answers.
> >
> > I think the comments on a sweet-spot for the initialization scale and tradeoff of encountering saddles in connection with the discussion on optimization speed (cf also point 3 of reviewer Bqc1) make the motivation more explicit and could inspire new research projects. With a cautionary foreword, I think it could be inserted in the paper but I understand you are reluctant due to the speculation involved.
> >
> > I think the edits made to the paper make sense, and regarding the practical applicability concerns my point of view is that theoretical analysis has to start where things are not known, there is potential long term valuable insight in this line of research and the paper is well written and does not overreach; therefore I maintain my good opinion of it and my positive score.

---

### Official Review · Reviewer_gV27 · 2025-11-08

**Soundness:** 2
**Presentation:** 3
**Contribution:** 2
**Rating:** 4
**Confidence:** 4

**Summary:**

This paper analyzes how gradient flow (GF) escapes the saddle point at the origin when training deep ReLU networks initialized with small weights. The authors characterize the optimal escape direction of GF and show that it exhibits a low-rank structure in the deeper layers, where the leading singular value of layer $ \\ell $ is at least $ \\ell^{1/4} $ larger than the other singular values combined. They further suggest that deep ReLU networks follow saddle-to-saddle dynamics, where the weights transition between saddles of progressively higher rank during training.

**Strengths:**

The paper studies an interesting topic of escaping saddle points. The problem setting and assumptions are clearly presented, and the authors support their analysis with illustrative examples and well-designed figures. Overall, the paper is clearly written and effectively communicates its main ideas.

**Weaknesses:**

Although the authors depicted a detailed picture of the dynamics of GD in the vicinity of the saddle point at the origin, I am still not convinced that this is an actual phenomenon for the following reasons.

**Initialization magnitude:**
The current initialization schemes avoid initialization at the plateau of origin. In other words, using the standard initializations, we would not encounter such a setting.

**GF vs. GD:**
The analysis presented in the paper applies to GF. However, these results cannot be translated directly to GD. Specifically, GD with non-vanishing step size exhibits richer dynamics with dynamical stability constraints along its trajectory (e.g., Edge of Stability). Moreover, the optimal escape speed should be adjusted according to the stability constraints.

**Optimial vs. typical escape direction:**
The presented results on low rank are about the optimal escape direction. However, there is no proof that GD (or GF) will follow this escape trajectory. It might be the case that the typical escape routes of GD are non-optimal. In this regard, I don't understand the claim of GD bias towards low rank solutions, since there is no guarantee that GD will follow this optimal direction.

**Effect on the fully trained model:**
The main result deals with the state of the model in the first stage of training. I am not sure what effect a low-rank model early in the training has on the fully trained model. I am not convinced by the claim of saddle-to-saddle dynamics without any proof.

**Questions:**

1. Please explain the relation between the optimal escape direction from the saddle at the origin and the actual dynamics of GD.
2. Can you give a way to convert the speed defined by GF to GD?
3. In line 189, it says that you consider the case of adaptive learning rate, where $ \\eta = \\| \\theta \\|^{2-L}  $. Wouldn't it be extremely large near the origin for deep networks?
4. In the equation of the escape time (line 234), the dependency of $  t(r\_1) -t\_0$ on $ r $ for deep networks, given by $ r\^{2-L} $, is strange. It predicts that crossing smaller thresholds $ r $ takes a longer time (since $ L > 2 $), which does not make sense. Is there a problem with this result?

---

> ### Author Response · Authors · 2025-11-20
>
> Thank you for your thoughtful review.
>
> We first want to make a high level point: the behavior of deep networks is extremely complex and there is little hope to build a complete theory for it at once. A sensible strategy is instead to search for settings where a specific phenomenon can be isolated and analyzed mathematically, with the hope to ultimately combine these ideas into a complete theory. The goal of this paper is to identify a mechanism for the emergence of low-rank structure in the weights of deep networks. Given it is the first paper studying this specific phenomenon, we focused on identifying the simplest model that can exhibit such non-trivial behavior.
>
> Initialization scale: We agree that out theoretical results can require extremely small initializationthat might not be practical. There are two generalizations that could increase its relevance:
> - We believe that the low-rank bias extends to more reasonable initializations scales, because it is not necessary to be exactly aligned with the optimal escape direction to get some low-rank bias (see e.g. Proposition 3.4 which only require a fast enough escape speed). In practice, it is probably optimal to initialize with a scale that is large enough to not get stuck at the origin saddle for too long, but small enough to still take advantage of the sparsity bias that the saddles implicitly provides. There is evidence that the impact of the saddle at the origin starts be felt as soon as the initialization becomes small enough to leave the NTK regime (Tu et al., 2024) and we have strong evidence that real-world networks are not in the NTK regime.
> - While this paper only studies the first saddle at the origin, we believe that the other saddles that GD encounters might be studied along similar lines. The phenomenon of plateaus in the training loss followed by sudden drop in loss, possibly related to the emergence of new capabilities of the network, have been observed by practitioners.
>
> GF vs GD: Note that the main novelty of our paper is a description of the optimal escape directions of the saddle and its low-rank bias, which relevant for both GD and GF. The only section that really relies on GF is Section 2 whose purpose is to provide a simple explanation for why GF is naturally attracted to these optimal escape directions, we used GF for simplicity of exposition, everything can be extended to GD. Previous work has studied convergence of GD to so-called KKT points which are closely related to the escape directions, e.g. Ziwei Ji, Matus Telgarsky, The implicit bias of gradient descent on nonseparable data.
>
> Optimal vs typical escape direction: Extending to more typical escape directions is indeed very interesting. Proposition 3.4 already shows that at any fast enough escape directions the majority of the layers will be approximately rank 1, so it is reasonable to expect a similar low-rank bias along typical escape directions. It is likely that the other results could be extended to locally optimal escape directions.
>
> After the first saddle: This does indeed require further work, but we are confident that it might be possible, as discussed in point (5) of Section 5: in linear networks one can easily show that if the parameters are balanced and low-rank then their rank cannot increase, and as an extension if  the parameters are approximately balanced/low-rank then their rank can only increase very slowly. This is the reason why we put emphasis on the fact that in our setting, the low-rank layers are also approximately linear: after the first saddle the low-rank layers will behave approximately like a linear network and so they will remain low-rank until the next saddle. There remains a lot of work to make this intuition rigorous, so we leave it future work.
>
>
> Regarding your questions:
> (1) We believe that the GF dynamics of section 2 and the split into radial and norm dynamics gives the most insight at the smallest complexity. The GD case with a small enough learning rate will then be an approximation of the GF dynamics, for more details we will add references to the aforementioned work of Ziwei Ji, Matus Telgarsky and many others that have studied a very similar technical question in much more details and generality.
>
> (2) These will be the same speed, with a norm scaling roughly as $(1+2 \eta s)^t$ instead of $\exp(2st)$ for $L=2$. For $L>2$ there might not be such a simple formula, but assuming a small enough $\eta$ it will approximate the explicit GF solution.
>
> (3) Yes, it would be infinitely large to compensate the gradient itself being infinitely small.
>
> (4) You are right, there is a simple sign mistake, it should be $\frac{1}{(L-2)Ls}[\|\theta(t_0)\|^{\,2-L} - r^{2-L}]$. Thank you for noticing this error.

---

### Meta-Review · Area_Chair_uJ2C · 2026-01-13

**Summary:**

The paper is clearly written and tackles an interesting theoretical question: how deep ReLU networks initialized at very small scale escape the origin saddle, and what structure optimal escape directions exhibit. Reviewers appreciated the clean problem formulation, the coherent separation of norm and directional dynamics used to build intuition, and the technically precise characterization of escape directions, including the depth-dependent low-rank bias that becomes stronger in deeper layers. The theoretical development is well organized, the main statements are easy to locate, and the paper includes concrete illustrative figures and a useful counterexample that clarifies that rank-one behavior is not universal, strengthening the conceptual message. The main concerns centered on relevance and scope rather than novelty of the mathematical direction. Reviewers questioned whether the tiny-initialization "plateau" regime is representative under standard practice, whether results stated and analyzed under gradient flow meaningfully translate to discrete gradient descent with nontrivial step sizes and stability constraints, and whether characterizing the optimal escape direction is informative about typical training trajectories. Several reviewers also emphasized that the empirical support is limited, largely to a single MNIST setup with carefully tuned hyperparameters, and that the broader "saddle-to-saddle dynamics" narrative remains conjectural with no formal proof beyond the first escape.

**Reviewer Concerns:**

The rebuttal addressed several points, but the most important concerns remain about practicality and empirical validation. On the positive side, the authors acknowledged that the experimental scope is narrow and committed to adding additional datasets (e.g., CIFAR) and clarifications; they also corrected a concrete technical mistake in the escape-time expression, which resolves that specific issue. They additionally clarified the intended role of gradient flow, arguing that the main novelty is the geometric characterization of escape directions and that the GF analysis is mainly explanatory, while discrete GD with sufficiently small step size should approximate the same behavior; they cited relevant prior literature (e.g., convergence to KKT-like points) to support this connection. This improves plausibility and clarity.

However, the central reviewer concerns are only partially addressed. First, the practicality concern is still outstanding: the paper does not convincingly demonstrate that the small-initialization plateau regime is representative of standard modern training (or that the effect persists in realistic regimes), beyond qualitative arguments. Second, the empirical evidence remains limited at decision time: the main demonstration is a single MNIST setup with carefully tuned hyperparameters and a toy projected-GD experiment, and there is no robust evaluation across datasets, architectures, or standard training choices. Third, the “optimal versus typical” gap is still not fully closed: while the rebuttal emphasizes a result that applies beyond the global optimum (fast-enough escapes imply many approximately rank-1 layers) and argues why typical trajectories may inherit similar bias, the paper still lacks a general guarantee, or broad empirical confirmation, that standard training reliably tracks near-optimal escape routes. Finally, the broader saddle-to-saddle dynamics narrative remains conjectural beyond the first escape; the rebuttal provides intuition and analogies (e.g., linear balanced low-rank dynamics) but does not supply proofs or strong empirical evidence that later-stage rank transitions occur as described.

**Reviewer Scores:**

Reviewer gV27 would likely remain near their original position, perhaps moving slightly upward after the correction of the sign error and the added clarification on how GF-based intuition relates to GD, but the major concerns about initialization relevance and the lack of a principled link from optimal escape directions to typical GD trajectories would still stand; I would expect at most a shift from 4 to 5, or staying at 4 with a more positive tone. Reviewer bYb4 already expressed a consistently positive view and reaffirmed it after the rebuttal; I would expect their score to remain at 8. Reviewer Bqc1 would likely remain around 6, as the rebuttal clarifies motivation and future directions and promises broader experiments, but does not fundamentally expand the empirical validation or prove the full saddle-to-saddle dynamics; I would not expect a large change. Reviewer Y4Ze would likely remain around 6 as well, since clarity improvements and planned extra experiments address part of their feedback, but the limited experimental scope and restrictive assumptions remain; at most a small upward adjustment in confidence rather than score.

---

### Decision · Program_Chairs · 2026-01-26

Accept (Poster)